# ACCELERATE AUTOREGRESSIVE NORMALIZING FLOWS SAMPLING WITH GS-JACOBI ITERATION

## ABSTRACT

AutoRegressive Normalizing Flows (abbreviated as AR Flow) enjoy extensive applications in tasks such as density estimation and image generation. However, due to the causal affine coupling blocks requiring sequential computation, the sampling process is extremely slow. In this paper, we demonstrate that through a series of optimization strategies, such AR Flows sampling can be greatly accelerated by using the Gauss-Seidel-Jacobi (abbreviated as GS-Jacobi) iteration method. Specifically, we find that blocks in AR Flows have varying importance: a small number of blocks play a major role in image generation, while other blocks contribute relatively little; some blocks are sensitive to initial values and prone to numerical overflow, while others are relatively robust. Based on these two characteristics, we propose the Convergence Ranking Metric (CRM) and the Initial Guessing Metric (IGM): CRM is used to identify whether a Flow block is "simple" (converges in few iterations) or "tough" (requires more iterations); IGM is used to evaluate whether the initial value of the iteration is good. The TarFlow was chosen as the main experimental subject in our study owing to its SOTA performance on several benchmarks. Experiments on four TarFlow models demonstrate that GS-Jacobi sampling can significantly enhance sampling efficiency while maintaining the quality of generated images (measured by FID), achieving speed-ups of 4.53× in Img128cond, 5.32× in AFHQ, 2.96× in Img64uncond, and 2.51× in Img64cond without degrading FID scores or sample quality.

## 1 INTRODUCTION

Image generation models have been widely applied in various scenarios. As an instance, normalizing-flow-based models, from the original NICE [11] model, to improved RealNVP [12] and Glow [21] models, offer unique advantages through their invertible architecture that applies sequence of lossless transformations to noise vectors, but show limited performance in the generation of high solution and complex images.

Recently, AutoRegressive Flow models, especially TarFlow [38] introduces stacks of autoregressive Transformer blocks (similar to MAF [27]) into the building of affine coupling blocks to do Non-Volume Preserving, combined with guidance [16] and denoising [5], finally achieves state-of-the-art results across multiple benchmarks. However, AR Flow's sampling efficiency suffers from a critical bottleneck: the causal structure within each affine coupling block forms a nonlinear RNN. This forces strictly sequential computation during sampling, where each step must wait for the full update of preceding series, resulting in significantly reduced computational efficiency for large-scale image generation tasks.

In this paper, we try to solve this. We first transform the nonlinear RNN in the sampling phase into a diagonalized nonlinear system, then we can employ iteration-based solvers such as Gauss-Seidel-Jacobi (GS-Jacobi) iteration [26] [33] [31]. However, naively applying GS-Jacobi iteration leads to generation failure (see 1st and 2nd row of Figure 4). Through detailed analysis, we discover that blocks in the AR Flow have varying importance: a small number of blocks play a major role in image generation tasks, while other blocks contribute relatively little; some blocks are sensitive to initial values and prone to numerical overflow, while others are relatively robust. Based on these two characteristics, we propose the Convergence Ranking Metric (CRM) and the Initial Guessing Metric (IGM): CRM is used to identify whether a AR Flow block is "simple" (converges in few iterations) or

"tough" (requires more iterations); IGM is used to evaluate whether the initial value of the iteration is appropriate. Leveraging these two metrics, we present a sampling algorithm that substantially reduces calculation amount required.

As a summary, we list our contributions as follows:

- Transform the sampling process of the AR Flow into a diagonalized nonlinear system, and apply a Gauss-Seidel-Jacobi hybrid iteration scheme to it, providing corresponding error propagation analysis and convergence guarantees;

- To identify the non-uniform transformation patterns in affine coupling blocks and control the number of iterations, we propose the Convergence Ranking Metric (CRM) to evaluate and measure them;

- To control the stability of iteration initial values to avoid numerical overflow, we propose the Initial Guessing Metric (IGM);

- Comprehensive experiments demonstrate 4.53× speedup in Img128cond, 5.32× in AFHQ, 2.96× in Img64uncond, 2.51× in Img64cond sampling without measurable degradation in FID scores or sample quality in TarFlow.

## 2 RELATED WORK

**Normalizing Flow Based Models**     In the field of image generation, numerous methods have been proposed. From PixelRNN ([35]), GANs [14], to DDPM [18], Stable Diffusion ([29]). Diffusion models seem to dominate this field, but normalizing flows still offer unique advantages, including exact invertibility [37] enabling precise density estimation [34], single-step sampling for efficient generation [15], and structured latent spaces that support interpretable manipulation [6].

Normalizing Flows learn an invertible model $f$ that transforms noise $z$ into data $x$, such that $x = f(z)$. The key to build invertible models is accessible inverse function with Jacobi determinant easy to calculate, series of flow models accomplishes this through coupling layers. NICE [11] introduced the additive coupling layers. To enhance the non-linear capability, RealNVP [12] integrated scaling and shifting to the non-volume preserving transform as the affined coupling layer. Glow [21] improved the images generation by introducing invertible $1 \times 1$ convolution, and Flow++ [17] included attention mechanic. The most significant advantage of these models is that the inverse function is explicit and Jacobi matrix is lower triangle. This can avoid the complex calculation in the general invertible ResNet framework proposed in [4]. Implicit Normalizing Flow [23] trys to eliminate the need for multi-layer stacking by applying deep equilibrium models [3; 36]. However, overly simple structure makes these flow models less nonlinear.

To improve this, normalizing flows are combined with autoregressive models. IAF [22] pioneered dimension-wise affine transformations conditioned on preceding dimensions to improve variational inference. MAF [27] utilized the MADE [13] to create invertible autoregressive mappings. NAF [19], which replaced MAF's affine transformations with per-dimension monotonic neural networks to enhance expressivity. T-NAF [28] augmented NAF by integrating a single autoregressive Transformer, whereas Block Neural Autoregressive Flow [9] adopted an end-to-end autoregressive monotonic network design. TarFlow [38] proposed a Transformer-based architecture together with a set of techniques to train high performance normalizing flow models and show SOTA in many fields, thus becomes the sampling object of this article.

**Parallel solving of linear/nonlinear systems**     Linear/nonlinear systems refer to $f_i(\mathbf{x}) = 0, \mathbf{x} \in \mathbb{R}^n$, where $f_i, i = 1, \ldots, n$ is a linear/nonlinear function. The parallel solution of these systems is an important problem in scientific computing. [30] established methods such as Jacobi, Gauss-Seidel, successive over-relaxation (SOR), and Krylov subspace techniques for linear systems. Block-Jacobi iterations [1; 2; 8] use GPU parallelization to solve linear/nonlinear equations. As a special case, when $f_i$ takes the form $f_i(\mathbf{x}) = \mathbf{x}_i - g_i(\mathbf{x}_{<i})$, it is called an autoregressive system. Lots of approaches have been proposed to accelerate autoregressive computation. [25] introduced probability density distillation for transferring knowledge from slow autoregressive models to faster computation. MintNet [32] developed a specialized Newton-Raphson-based fixed-point iteration method to speed up autoregressive inversion. Similar theoretical concepts were earlier explored by [24] without empirical validation.

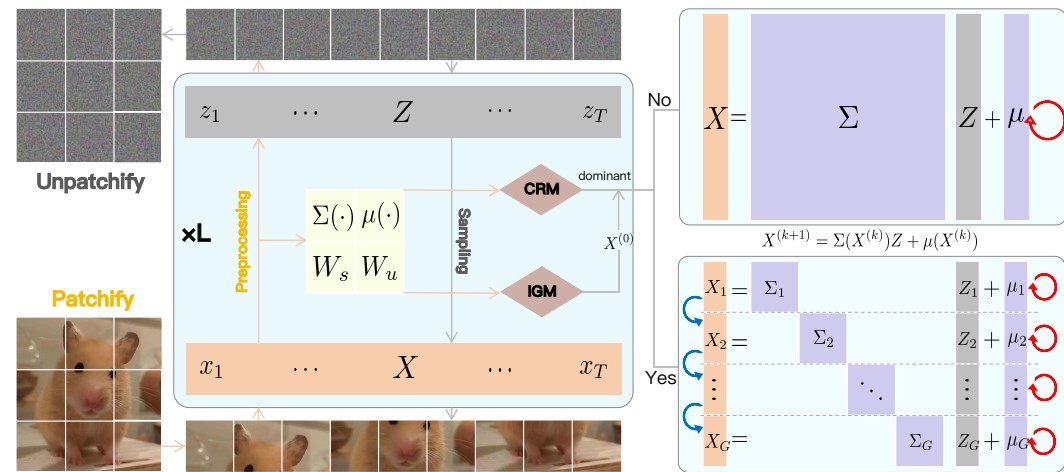

Figure 1: Simple intuition diagram of GS-Jacobi sampling. First pass forward a small batch of images to compute Initial Guessing Metric (IGM) and Convergence Ranking Metric (CRM) for each block. When sampling, the initial iteration value $X^{(0)}$ is determined by IGM; for blocks whose CRM is non-dominant, parallel Jacobi iterate $X$; for CRM-dominant blocks, segment $X$ into small modules $X_g$, parallel Jacobi iterating within modules, then serially deliver to next module.

## 3 METHODS

### 3.1 JACOBI MODE FIXED POINT ITERATION SAMPLING

Let $z$ denotes the noise direction and $x$ denotes the image direction, both with size $(B, T, C)$, where B,T,C represent batch size, patchified sequence length, and feature dimension, respectively. For AR Flow models, an affine coupling block can be written as:

$$\text{Forward:} \quad z_t = \exp(-s(x_{<t}))(x_t - u(x_{<t})), \quad \text{Inverse:} \quad x_t = \exp(s(x_{<t}))z_t + u(x_{<t}). \quad (1)$$

for $t = 1, \cdots, T$ and $x_1 = z_1$. $x_{<t} := \{x_i\}_{i=1}^{t-1}$ denotes the history before time $t$, $s(x_{<t}), u(x_{<t})$ generated from some deep neural network (IAF, MAF use MADE[13], TarFlow use Attention). In forward direction, all $x_t$ are given, so all $s(x_{<t}), u(x_{<t})$ can be calculate in parallel. But in inverse direction, $x_t$ can only be computed serially after $x_{<t}$ has been solved. In Table 3a, it takes about 213 seconds for such serial sampling to generate 100 128×128 images with a single A800 GPU.

Denote $\exp(-s(x_{<t})) = \sigma_t^{-1}, u(x_{<t}) = u_t$, the former process can be written in a matrix form, and with $\sigma_1 = 1, u_1 = 0$, then the transform from $X$ to $Z$ can be seen as an non-linear system:

$$\text{Forward:} \quad Z = \Sigma^{-1}(X)(X - \mu(X)), \quad \text{Inverse:} \quad X = \Sigma(X)Z + \mu(X). \quad (2)$$

For the inverse process, we can view the target $X^*$ as the fixed point of the nonlinear system $g(X) = \Sigma^{-1}(X)Z + \mu(X)$, and then solve it using the non-linear Jacobi iteration [20]:

$$X^{(k+1)} = \Sigma(X^{(k)})Z + \mu(X^{(k)}), \quad x_t^{(k+1)} = \sigma_t^{(k)}z_t + u_t^{(k)} \quad (3)$$

parallel for $t = 1, \ldots, T$ with an initialized $X^{(0)}$. We propose Proposition (1) to explain the convergence and error propagation of Jacobi mode iteration under this nonlinear system. See Appendix A for detailed discussion.

**Proposition 1** (Converge and Error Propagation). *For fixed point iteration (3), let $\varepsilon^{(k)} = X^{(k)} - X^*$ be the error after $k$ iteration, $e_t$ be its $t$-th component, $f_t^{(k)} = x_t^{(k)} - \sigma_t^{(k)}z_t - u_t^{(k)}$, then:*

- *Equation (3) converges strictly after $T - 1$ times,*

- $e_t^{(k)} \approx -\sum_{i=k+1}^{t-1} \gamma_{ti}^{(k)} e_i^{(k)}$, *with* $\gamma_{ti}^{(k)} = \frac{\partial f_t}{\partial x_i}\big|_{X^{(k)}}, t \geq k + 2.$

This iteration method involves two components: the initial value $X^{(0)}$ and the maximum number of iterations. As shown in Figure 4, different initialization strategies lead to different convergence

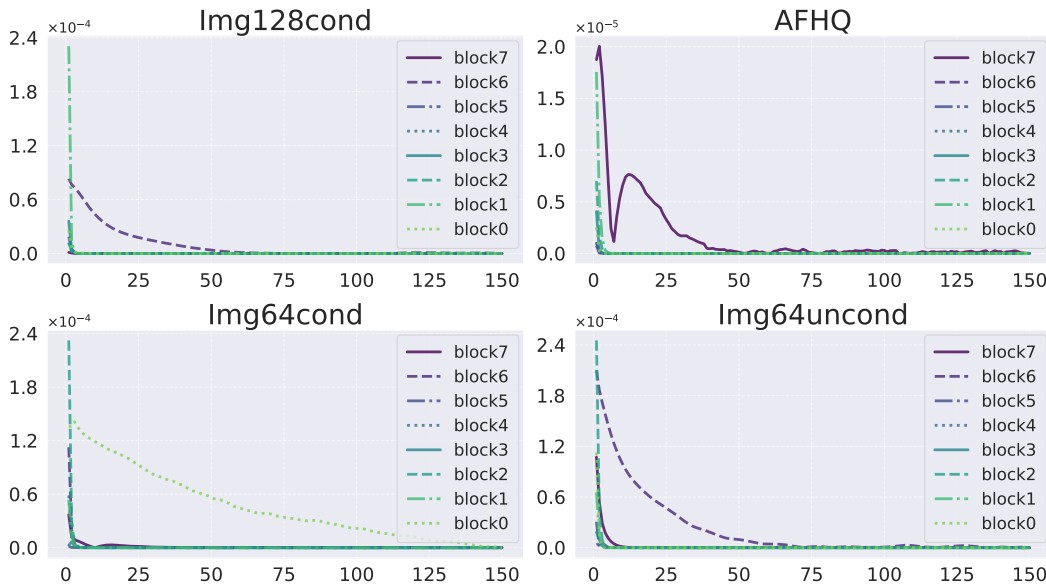

Figure 2: The distance between $X^{(k)}$ (up to 150 times) and target $X^*$ of all 8 blocks in four models. Most blocks converge within iteration times $<< T$, with each model exhibiting only one or two slowly descending curves.

effects, with poor strategies causing model collapse. Also as shown in Figure 2, different blocks converge at varying speeds, some blocks converge quickly while others slowly, suggesting that we should employ different iteration strategies for different blocks. We propose Initial Guessing Metric and Convergence Ranking Metric to address these two issues, respectively.

## 3.2 INITIAL GUESSING METRIC

A common choice for initialization is to take $X^{(0)} = Z = [z_1, z_2, \ldots, z_t]'$, i.e, the output of the former block, with intuition Flow models transform images "gradually" [38], which means that the difference between adjacent affine coupling blocks maintain stable. As shown in Figure 3, the change from noise to image in most steps is gradual, and $Z$ locates in the neighbor of $X^*$, which can be a good initial guessing. However, in practice, we find that take all $X^{(0)} = Z$ cause numeric collapse in Img64cond models in Block0, as shown in the 1st row of Figure 4.

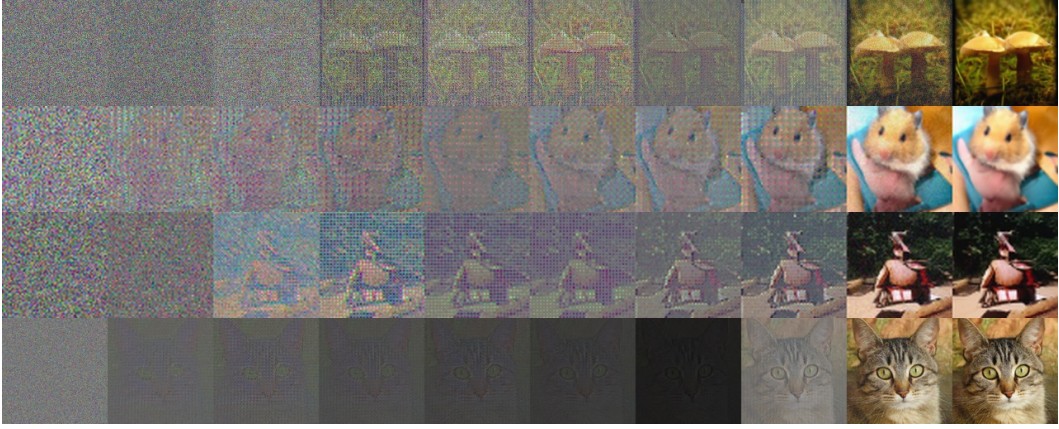

Figure 3: The trace of the sampling in four models. From top to bottom: Img128cond, Img64cond, Img64uncond, AFHQ. From left to right: noise, Block 7-0, denoised image.

An alternative workable guessing is $X^{(0)} = Z_0 = [z_1, 0, \ldots, 0]'$, since pixel value ranges from -1 to 1 and centers in 0. A natural strategy is comparing $Z$ and $Z_0$ and choose the better one. Since the

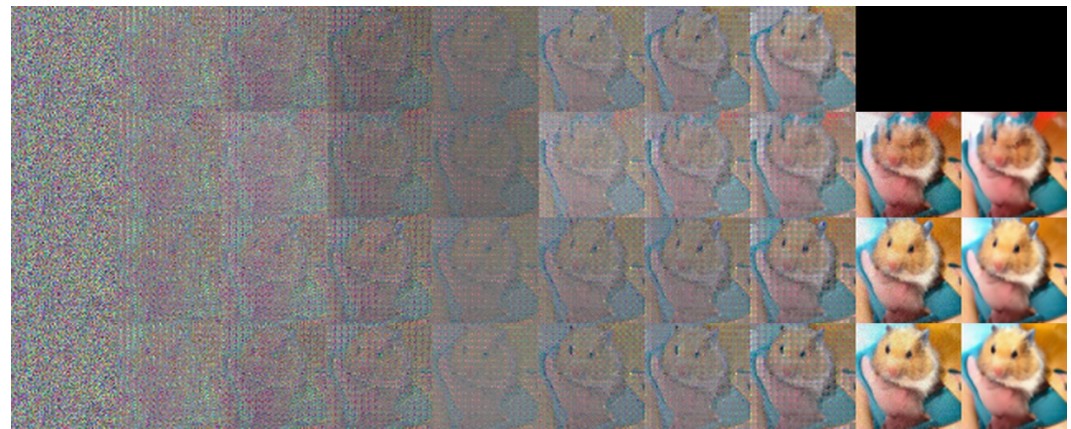

Figure 4: The influence of different initial value and iteration times of an Img64cond sample. From top to bottom: Set all $X^{(0)} = Z$, Jacobi 30 times; Adaptive initialized according to IGM, Jacobi 20 times; Adaptive by IGM, Jacobi 30 times; GS-Jacobi [0/7-16/8-10/13-6]
.

worst inflation occurs at first few iteration, we define following "Initial Guessing Metric":

$$\text{IGM}(X^{(0)}) = ||\Sigma(X^{(0)})Z + \mu(X^{(0)}) - X^*||_2. \tag{4}$$

to measure a rough distance with $X^{(0)}$ chosen from $\{Z, Z_0\}$. In Appendix D, different norms showed similar results, spectral norm is a little better and we use it in this paper. We can calculate IGM with the following steps:

- Select a batch of images from the training set, patching to size $(B, T, C)$, which is $X^*$;
- Forward passing $X^*$ through AR Flow blocks to get $Z = \Sigma^{-1}(X^*)(X^* - \mu(X^*))$;
- Calculate the residual $\Sigma(X^{(0)})Z + \mu(X^{(0)}) - X^*$ with both $\{Z, Z_0\}$;
- Calculate the mean of residual in the dim $B$, calculate the norm of the $(T, C)$ matrix.

### 3.3 Convergence Ranking Metric

When sampling with (3), although all blocks converge strictly, some get nice solution with very small $k$, while others need $k$ near $T - 1$. As shown in Figure 2, Block6 of Img128cond, Block7 of AFHQ, Block0 of Img64cond, Block6 of Img64uncond behave worse compared to other blocks. To measure this difference, we propose the following Convergence Ranking Metric:

$$\text{CRM} = ||\Sigma^{-1}(X)X||_2||W_s||_2 + ||W_u||_2 \tag{5}$$

with $W_s, W_u$ the weight matrix of the project out layer (the final full connect layer) of $s(x_{<t}), u(x_{<t})$. $W_s$ measures the change of variance; $W_u$ measures the mean, and $\Sigma^{-1}(X)X$ measures the non-volume-preserving. See Appendix B for detailed derivation. CRMs can be calculated with the following steps:

- Extract the project out parameters for each AR Flow block, calculate $||W_s||_2, ||W_u||_2$;
- Select a small batch of images from the training set, go through the forward process, get $\Sigma^{-1}(X)X$ with size $(B, T, C)$, take means over $B$ dimension to get $(T, C)$ size matrixs;
- Calculate $||\Sigma^{-1}(X)X||_2$ then CRM for each block.

This metric doesn't strictly measure the convergence rate, only represents the relative convergence ranking among TarFlow blocks, therefore we call it a ranking metric. In Appendix D, different matrix norms behave similarly in relatively ranking, and we use spectral norm.

By CRM, we can know whether a block can converge rapidly or slowly, thus roughly determine the iteration times of (3). Blocks with dominant CRM values in Table 2 converge slowly in Figure

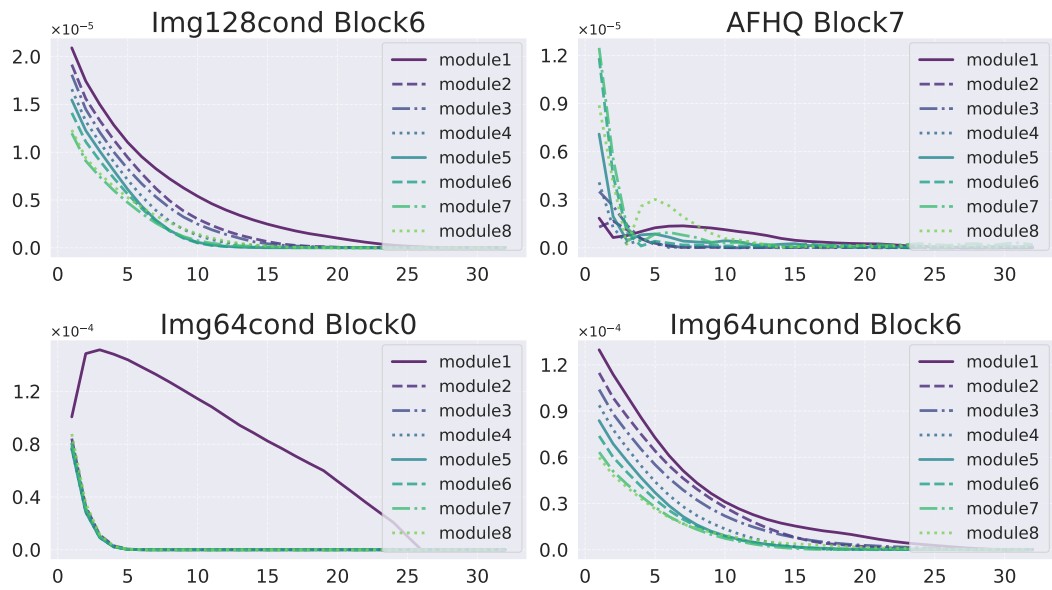

Figure 5: The distance between GS-Jacobi iteration and target $X_g^*$ of four tough blocks. All modules tend to converge within 30 iterations, and the 1st module suffer a more difficult trace.

2. In practice, although only very few blocks in TarFlow converge slowly, this severely affects the speed and effectiveness of the Jacobi iteration method: For "tough" blocks, fewer iterations result in poor generation quality (see 2nd row of Figure 4), while more iterations improve the model but simultaneously lose the speed advantage. As shown in Table 3a 3b, the Jacobi-30 strategy exhibits significantly inferior performance, while Jacobi-60 shows measurable improvement with much more time cost.

Although computed from a small sample set, we posit that IGM and CRM are intrinsic properties of the trained model, remaining fixed across different sample categories. These metrics do not require recomputation during sampling. Experimental validation of this is provided in Appendix E.

### 3.4 MODULAR GUASS-SEIDEL-JACOBI ITERATION

For a $(B, T, C)$ tensor, (3) updates all $T$ units in parallel, while "For" iteration updates 1 unit a time, serially run $T - 1$ times. Naturally, an in-between method is to update a set of units in parallel (with Jacobi) in one iteration, and serially go to another set, that's so-called Guass-Seidel-Jacobi iteration. Let $X := \{x_t\}_{t=1}^T$, $\{\mathcal{G}_g\}_{g=1}^G$ an non-decrease segmentation for time-step index $1 : T$, $X_g := \{x_t | t \in \mathcal{G}_g\}_{g=1}^G$, $X_{:g} := \bigcup_{i=1}^g X_i$, and similar defination for $\{Z, z_t\}, \{\Sigma, \sigma_t\}, \{\mu, u_t\}$. Then the concept of modular GS-Jacobi method can be shown in Figure 1, and detailed algorithm is shown in Appendix F.

All the analysis of Jacobi mode iteration is applicable to the modules of GS-Jacobi sampling. We point out that GS-Jacobi can effectively improve the solution for blocks with large CRM:

- The probability of numerical overflow due to initial guessing value is greatly reduced. The size of error matrix (6) is smaller thus the error cumsum (7) reduced;

- The convergence of each sub-Jacobi will be accelerated, since the modules closer to the back will have a more accurate initial value;

- An appropriate GS-Jacobi strategy (select $\mathcal{G}_g$ and maximum Jacobi iteration times) can achieve both accurate and fast solution.

We segment the tough blocks into 8 equal modules and apply GS-Jacobi iteration in Figure 5. In Figure 2, pure Jacobi iteration requires between 50 to 150 times to converge for tough blocks, whereas in Figure 5, the GS-Jacobi method reduces this number to approximately 30, and usually only module1 suffer a more difficult trace.

Table 1: Initial Guessing Metric for initialization with $Z, Z_0$ for four models.

| Blocks | Img s128cond | | Img64cond | | Img64uncond | | AFHQ | |
| $X^{(0)}$ | $Z$ | $Z_0$ | $Z$ | $Z_0$ | $Z$ | $Z_0$ | $Z$ | $Z_0$ |
|---|---|---|---|---|---|---|---|---|
| Block0 | 12.06 | 14.15 | $5.4 \times 1e5$ | 9.89 | 1443.13 | 9.98 | 21.81 | 26.28 |
| Block1 | 14.19 | 3.55 | 7.23 | 6.92 | 10.49 | 5.64 | 12.48 | 9.81 |
| Block2 | 3.35 | 5.66 | 6.19 | 5.92 | 5.89 | 6.25 | 11.88 | 8.73 |
| Block3 | 10.33 | 14.23 | 4.99 | 6.40 | 6.65 | 3.50 | 34.36 | 40.80 |
| Block4 | 9.04 | 29.26 | 10.64 | 13.45 | 3.13 | 8.47 | 15.58 | 46.59 |
| Block5 | 14.78 | 26.89 | 5.55 | 24.23 | 2.74 | 5.46 | 28.61 | 48.51 |
| Block6 | 53.42 | 42.03 | 4.31 | 20.18 | 34.98 | 5.64 | 13.91 | 51.48 |
| Block7 | 11.00 | 39.67 | 23.64 | 13.92 | 23.11 | 31.86 | 124.80 | 134.86 |

So, a proper strategy can take advantage of such modular iteration method. Ideally, IGM and CRM should be calculated for each GS-Jacobi modules to judge it is tough or not. Then for every modules, allocate more iteration to large CRM and vice versa. This can be seen as an adaptive strategy.

In practice, equal-size segmentation and same Jacobi times is usually enough. Then strategies can be denoted in the format [Stack-GS-J-Else]. Stack indicates the tough blocks should be segmented; GS indicates the number of equal size segmentation with length $T//\text{GS}$; J indicates the maximum Jacobi times of each module; Else indicates the maximum Jacobi times for other blocks with small CRM.

To determine the stacked blocks, select blocks with large CRM one by one until there are no dominant blocks in the remaining set. By Table 2, we stack Block6 in Img128cond, Block7 in AFHQ, Block0&6 in Img64uncond, Block0&7 in Img64cond.

## 4 EXPERIMENT

We train four models given by [38]: TARFLOW [4-1024-8-8-$\mathcal{N}(0, 0.15^2)$] for Conditional ImageNet 128×128 [10]; TARFLOW [8-768-8-8-$\mathcal{N}(0, 0.07^2)$] for AFHQ 256×256 [7]; TARFLOW [2-768-8-8-$\mathcal{N}(0, 0.05^2)$] for Unconditional ImageNet 64×64 [35]; TARFLOW [4-1024-8-8-$\mathcal{N}(0, 0.05^2)$] for Conditional ImageNet 64×64. The first three $T = 1024$, the last $T = 256$, and all four models have 8 TarFlow blocks. For convenience we will refer to them as Img128cond, AFHQ, Img64uncond and Img64cond.

### 4.1 INITIAL GUESSING METRIC

We first calculate IGMs for four models with 128 training images, as shown in Table 1. We find that there are not significant difference between two initializations in Img128cond and AFHQ, while the sampling of Img64cond and Img64uncond will collapse if initialize $X^{(0)} = Z$ for all blocks. This is evident in Table 1 since IGMs of Block0 in Img64cond and Img64uncond are pathological with $X^{(0)} = Z$, while set $X^{(0)} = Z_0$ can release this.

As shown in Figure 2, IGM is highly correlated with the potential maximum value occur during the iteration. We find that Img64cond and Img64uncond are more sensitive to the initial value. This may be because low-resolution images are more prone to mutations between pixels, which causes huge fluctuations in the attention layers parameters. In practice, the GS-Jacobi segmentation can greatly improve the problem of numerical overflow, so it is sufficient to simple initialize with $Z, Z_0$ by IGM.

### 4.2 CONVERGENCE RANKING METRIC

We calculate CRMs for four models with 128 images in Table 2 the same time with IGMs. Detailed components are shown in Appendix C. Table 2 is consistent with Figure 2, following the simpe rule: The larger the CRM, the more Jacobi times required for convergence, and vice versa.

An important property is, only very few blocks in a TarFlow model have relative large CRM. This may be because TarFlow, or other normalizing-flow based generative models are over-determined, which means that the amount of parameters is redundant relative to the generative capacity, and many blocks

Table 2: Convergence Ranking Metric of four TarFlow models, with dominant blocks bolded.

| Models | Img128cond | | AFHQ | | Img64uncond | | Img64cond | |
| --- | --- | --- | --- | --- | --- | --- | --- | --- |
| | CRM | Percent | CRM | Percent | CRM | Percent | CRM | Percent |
| Block0 | 6.52 | 5.22 | 51.85 | 6.33 | 22.29 | **50.71** | 141.22 | **74.58** |
| Block1 | 7.03 | 5.63 | 51.45 | 6.28 | 1.06 | 2.42 | 9.25 | 4.88 |
| Block2 | 3.08 | 2.47 | 66.76 | 8.16 | 1.01 | 2.29 | 1.36 | 0.72 |
| Block3 | 13.63 | 10.93 | 64.98 | 7.94 | 1.48 | 3.38 | 1.82 | 0.96 |
| Block4 | 9.66 | 7.74 | 73.77 | 9.01 | 0.77 | 1.77 | 7.68 | 4.05 |
| Block5 | 9.17 | 7.35 | 84.05 | 10.27 | 0.58 | 1.33 | 5.08 | 2.68 |
| Block6 | 70.54 | **56.57** | 76.64 | 9.36 | 14.78 | **33.62** | 3.08 | 1.62 |
| Block7 | 5.05 | 4.05 | 348.51 | **42.60** | 1.95 | 4.44 | 19.81 | **10.46** |

Table 3: FIDS of different GS-Jacobi strategies for four Models, with relative error <1% bolded.

| Strategy | FID (rel) | time (rate) | Strategy | FID (rel) | time (rate) |
| --- | --- | --- | --- | --- | --- |
| Original | 5.06 | 133.19 (1.00) | Original | 14.67 | 109.05 (1.00) |
| Jacobi-30 | 10.36 | 58.16 (2.29) | Jacobi-30 | 25.66 | 45.60 (2.39) |
| Jacobi-60 | 6.07 | 114.24 (1.17) | Jacobi-60 | 17.98 | 92.06 (1.18) |
| [6-1024-1-8] | 5.07 (0.20) | 33.11 (**4.02**) | [0/6-1024-1-10] | 15.27 (4.10) | 38.17 (2.86) |
| [6-1024-1-10] | 5.04 (0.00) | 36.31 (**3.67**) | [0/6-1024-1-20] | 14.77 (0.68) | 47.65 (**2.29**) |
| [6-1024-1-20] | 5.04 (0.00) | 52.98 (**2.51**) | [0/6-1024-1-30] | 14.72 (0.34) | 57.16 (**1.91**) |
| [6-1-128-10] | 5.50 (8.70) | 48.46 (2.75) | [0/6-2-64-20] | 15.22 (3.7) | 51.11 (2.13) |
| [6-2-64-10] | 5.19 (2.60) | 34.08 (3.91) | [0/6-4-32-20] | 15.18 (3.5) | 36.89 (2.96) |
| [6-4-32-10] | 5.22 (3.20) | 27.59 (4.83) | [0/6-8-16-20] | 16.44 (12.1) | 28.11 (3.88) |
| [6-8-16-10] | 5.40 (6.72) | 24.37 (5.46) | [0/6-16-8-20] | 21.16 (44.2) | 26.14 (4.17) |
| [6-1-256-10] | 5.30 (4.74) | 78.93 (1.69) | [0/6-2-128-20] | 15.03 (2.50) | 76.41 (1.43) |
| [6-2-128-10] | 5.10 (0.79) | 48.75 (**2.73**) | [0/6-4-64-20] | 14.81 (0.95) | 50.06 (**2.18**) |
| [6-4-64-10] | 5.05 (0.00) | 35.64 (**3.74**) | [0/6-8-32-20] | 14.80 (0.89) | 36.84 (**2.96**) |
| [6-8-32-10] | 5.09 (0.59) | 29.41 (**4.53**) | [0/6-16-16-20] | 15.17 (3.40) | 31.38 (3.47) |
| [6-16-16-10] | 5.16 (2.00) | 26.38 (5.05) | [0/6-32-8-20] | 17.47 (19.0) | 29.86 (3.65) |
| (a) Img128cond with cfg=1.5 lr=0.97 | | | (b) Img64uncond with cfg=0.2 attn=0.3 lr=0.9 | | |

don't modify the images drastically, only carefully crafted. As shown in Figure 3, visually, many middle blocks have no obvious changes, which provides the possibility of GS-Jacobi acceleration.

To identify such "tough" blocks, we just need to repeatedly select the block with the largest CRM until there is no dominant block in the remaining blocks. So for Img64cond, we first select Block0, but Block7 with CRM 10.46 is still dominant in remaining, so Block7 is included.

### 4.3 QUANTITATIVE EVALUATIONS WITH FID

We tune the hyperparameters cfg (classifier free guidance), lr (denoise learing rate), attntemp (attention temperature), sampling 50000 images with "For" iteration to restore the FIDs results in [38]. Treat it as the target FIDs, then keep the hyperparameters consistent, sampling with different GS-Jacobi strategies, recording the FIDs, relative error (%), running time (100 s) and accelerating rate. Rates with a relative error less than 1% are bolded. The strategy is as stated above [Stack-GS-J-Else]. All samplings are performed on 8 A800 GPUs with 80G memory.

In Table 3a Img128, keep "For" iteration for tough Block6, just few pure Jacobi for other blocks are enough to get good FID, like [6-1024-1-10], speeds up 3.67×. Then we fixed the total Jacobi times for Block6 with 128 and 256, and try different [GS-J] pairs. We found that simple strategies, like [6-8-32-10] can achieve results with relative error < 1% and surprising speed-up.

Similar results occurred in the sampling for AFHQ, as shown in Table 4a. Since the two models both have just one tough block, the acceleration rate behave similarly. For Img64 models, the situations

Table 4: FIDS of different GS-Jacobi strategies for four Models, with relative error <1% bolded.

| Strategy | FID (rel) | time (rate) | Strategy | FID (rel) | time (rate) |
|---|---|---|---|---|---|
| original | 13.60 | 109.24 (1.00) | Original | 4.42 | 12.16 (1.00) |
| [7-1024-1-10] | 13.61 (0.07) | 26.58 (**4.11**) | [0/7-256-1-6] | 4.42 (0.00) | 5.26 (**2.31**) |
| [7-1024-1-20] | 13.60 (0.00) | 37.69 (**2.90**) | [0/7-256-1-8] | 4.42 (0.00) | 5.81 (**2.09**) |
| [7-1024-1-30] | 13.60 (0.00) | 48.70 (**2.24**) | [0/7-256-1-10] | 4.42 (0.00) | 6.37 (**1.91**) |
| [7-1-128-10] | 14.70 (8.08) | 33.36 (3.27) | [0/7-256/1-1/64-6] | 4.41 (0.00) | 6.78 (**1.79**) |
| [7-2-64-10] | 14.27 (4.92) | 23.56 (4.54) | [0/7-256/2-1/32-6] | 4.40 (0.00) | 5.51 (**2.20**) |
| [7-4-32-10] | 14.15 (4.04) | 19.06 (5.73) | [0/7-256/4-1/16-6] | 4.54 (2.71 | 4.86 (2.50) |
| [7-8-16-10] | 15.52 (14.1) | 16.83 (6.49) | [0/7-256/8-1/8-6] | 5.35 (21.0) | 4.59 (2.65) |
| [7-1-256-10] | 14.21 (4.48) | 53.62 (2.04) | [0/7-256/4-1/24-6] | 4.38 (0.00) | 5.34 (**2.28**) |
| [7-2-128-10] | 14.07 (3.45) | 33.98 (3.21) | [0/7-256/8-1/13-6] | 4.41 (0.00) | 5.03 (**2.42**) |
| [7-4-64-10] | 13.82 (1.62) | 24.81 (4.40) | [0/7-16/8-8/13-6] | 4.50 (1.81) | 4.63 (2.63) |
| [7-8-32-10] | 13.73 (0.96) | 20.54 (**5.32**) | [0/7-16/8-10/13-6] | 4.43 (0.23) | 4.85 (**2.51**) |
| [7-16-16-10] | 14.12 (3.82) | 18.49 (5.91) | [0/7-16/8-12/13-6] | 4.42 (0.00) | 4.97 (**2.45**) |
| (a) AFHQ with cfg=3.4 lr=1.4 | | | (b) Img64cond with cfg=2.0 lr=1.0 | | |

are quite different. As shown in Table 3b Img64uncond, acceleration rates are not as high as single tough block models because it stacks both Block0 and 6, but still speeds up about 3×.

In Img64uncond, we treat two tough blocks equal since the CRMs have no absolute gap. For Img64cond, we first stack both Block0 and Block7 to original "For" loop, get the rate 2.31. Then we keep Block0 unchanged, try different strategies for Block7, the rate can be improved to 2.42. From Figure 5 and Table 2, we notice that Block0 behaves much tougher than any other blocks, so we segment Block0 into more modules and get 2.51× speed up.

Based on all above experiments, we can conclude that, the fewer the blocks with dominant CRMs and the longer the time step after patching, the more significant the acceleration can achieve by GS-Jacobi sampling. This is consistent with intuition. GS-Jacobi achieves acceleration by iterating batches of equations in parallel, avoiding repeated serial updates of the kv caches in the "For" loop.

### 4.4 OTHER FLOW MODELS

To demonstrate the versatility of the GS-Jacobi iteration across AR Flows, we also do acceleration for Masked Autoregressive Flow (MAF[27]) on the MNIST dataset with following steps: Train a MAF for MNIST with 5 maf_layer; Forward 50,000 images $X^*$ into MAF, get normal noise $Z$; Backward $Z$ with inversed MAF to regenerate $X$, with both baseline "For" loop and pure Jacobi iteration; Record the sampling time (in seconds), and MAE (for per pixel) between $X, X^*$. The results are shown in Table 5.

## 5 CONCLUSION

In this paper, we comprehensively optimize the sampling process of Autoregressive Flow models. By identifying the non-uniform transformation patterns across affine coupling blocks and proposing IGM and CRM, we effectively address the problems of initial value chosen and convergence rate differences. The introduction of the GS-Jacobi iteration and its in-depth error propagation analysis provides practical and efficient solution for TarFlow sampling. The experimental results on multiple TarFlow models show the superiority of proposed methods. The GS-Jacobi sampling achieving speed-ups of 4.53× in Img128cond, 5.32× in AFHQ, 2.96×

| Number of iteration | Time (s) | Speed up ratio | MAE |
|---|---|---|---|
| baseline | 4903 | 1.00 | 0.000 |
| 10 | 563 | 8.73 | 0.043 |
| 20 | 772 | 6.35 | 0.019 |
| 30 | 1114 | 4.40 | 0.007 |

Table 5: Performance metrics of acceleration for MAF. In first column, baseline is original "For" loop, others are Jacobi iteration with different time.

in Img64uncond, and 2.51× in Img64cond without degrading sample quality, which is of great significance for the application of TarFlow models.

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

## A CONVERGENCE AND ERROR PROPAGATION

Indeed, (3) is an equivalent form of the diagonal Newton method. Let $f(X) = X - g(X)$, to find its root, the iteration of diagonal Newton method is:

$$X^{(k+1)} = X^{(k)} - D_f^{-1}(X^{(k)})f(X^{(k)})$$

with $D_f^{-1}(X^{(k)})$ the diagonal of Jacobi matrix $J_f(X) = \partial f / \partial X$. Because $g(X)$ is causal designed in $T$ dimension, $D_f^{-1}(X^{(k)}) = I$. Then the iteration formula of diagonal Newton method is again (3).

Since $I - D_f^{-1} J_f$ is a strictly lower triangle matrix with zero spectral norm, the fixed point iteration (diagonal Newton method) is superlinear convergence in the neighbor of $X^*$. Indeed, this iteration can converge strictly after $T - 1$ times:

$$x_t^{(k+1)} - x_t^{(k)} = (\sigma_t^{(k)} - \sigma_t^{(k-1)})z_t + (u_t^{(k)} - u_t^{(k-1)})$$

Initially, $x_1^* = z_1$, for $t = 2$, one iteration can get the accurate $x_2^*$, and so for $x_t^*$. But we don't need a absolutely accurate solution, since the difference between $x_t^{(k)}$ and $x_t^*$ reduced after each iteration. Let $\varepsilon^{(k)} = X^{(k)} - X^*$ be the error after $k$ iteration, $f_t^{(k)} = x_t^{(k)} - \sigma_t^{(k)} z_t - u_t^{(k)}$, for $t$-th component:

$$e_t^{(k+1)} = e_t^{(k)} - f_t^{(k)}$$

$$\approx e_t^{(k)} - \sum_{i=1}^{t} \frac{\partial f_t}{\partial x_i}\Big|_{X^{(k)}} e_i^{(k)}$$

$$= -\sum_{i=1}^{t-1} \frac{\partial f_t}{\partial x_i}\Big|_{X^{(k)}} e_i^{(k)}$$

with the $\approx$ obtained by first-order Taylor expansion at $X^*$. Denote $\frac{\partial f_t}{\partial x_i}\Big|_{X^{(k)}}$ as $\gamma_{ti}^{(k)}$, then $e_t^{(k+1)} \approx -\sum_{i=1}^{t-1} \gamma_{ti}^{(k)} e_i^{(k)}$, and the error recursion can be written in a matrix form:

$$\varepsilon^{(k+1)} = \Gamma^{(k)}\varepsilon^{(k)}, \Gamma^{(k)} = -\begin{bmatrix} 0 & & & \\ \gamma_{21} & 0 & & \\ \vdots & \ddots & \ddots & \\ \gamma_{T1} & \cdots & \gamma_{T,T-1} & 0 \end{bmatrix}^{(k)}. \tag{6}$$

Since $\varepsilon^{(k)} = \prod_{j=0}^{k-1} \Gamma^{(j)} e^{(0)}$ and $e_1^{(0)} = 0$, the product of $\Gamma^{(k)}$ will move down the lower triangle part one unit afer each iteration, so the error will go to 0 after $T - 1$ iteration.

Thus for each component $e_t$:

$$e_t^{(k)} = -\sum_{i=k+1}^{t-1} \gamma_{ti}^{(k)} e_i^{(k)} \tag{7}$$

## B   DERIVATION FOR CONVERGENCE RANKING METRIC

To measure the convergence difference between affine coupling blocks, a direct method is to calculate the norm of error recursion matrix (6). But, analytically,

$$
\begin{aligned}
\gamma_{ti} &= -\frac{\partial \sigma_t}{\partial x_i} z_t - \frac{\partial u_t}{\partial x_i} \\
&= -\sigma_t z_t \frac{\partial s(x_{<t})}{\partial x_i} - \frac{\partial u(x_{<t})}{\partial x_i}
\end{aligned}
$$

the derivative of $\sigma_t, u_t$ is hard to calculate since they are generated by series of attention layers, thus we propose a simple but vaild alternatives. Since $s(x_{<t}), u(x_{<t})$ consist of series of deep neural network and a project out layer:

$$
s(x_{<t}) = W_s \text{NN}(x_{<t}) + b_s \\
u(x_{<t}) = W_u \text{NN}(x_{<t}) + b_u
$$

then:

$$
\gamma_{ti} = -(\sigma_t z_t W_s + W_u) \frac{\partial \text{NN}(x_{<t})}{\partial x_i}
$$
$$
\Gamma = -(\Sigma(X) Z W_s + W_u) J(X)
$$

Since the norm of Jacobi matrix of attention layers behave in coordination with the previous item, which can be dropped out without effect in ranking.

In practice, we can replace the non-volume-projection parts with $\Sigma^{-1}(X)X$ to calculate it at the same time with IGM in the forward direction. Intuitively $\Sigma(X)Z$ and $\Sigma^{-1}(X)X$ measure the same property from symmetric direction. Then the simplified norm of error can be written as:

$$
\text{CRM} = ||\Sigma^{-1}(X)X||_2 ||W_s||_2 + ||W_u||_2
$$

## C COMPONENTS OF CONVERGENCE RANKING METRIC

The components of CRM for Non-Volume-Preserving $||\Sigma^{-1}X||$, variance $||W_s||$ and mean $||W_u||$ are shown as Table 6.

Table 6: Components of Convergence Ranking Norm, with dominant CRMs bolded.

| | Imagenet128 | | | | AFHQ | | | |
|---|---|---|---|---|---|---|---|---|
| | $||\Sigma^{-1}X||$ | $||W_s||$ | $||W_u||$ | CRM | $||\Sigma^{-1}X||$ | $||W_s||$ | $||W_u||$ | CRM |
| Block0 | 20.06 | 0.31 | 0.24 | 6.52 | 52.78 | 0.96 | 0.86 | 51.85 |
| Block1 | 14.77 | 0.46 | 0.18 | 7.03 | 39.32 | 1.28 | 0.75 | 51.45 |
| Block2 | 13.65 | 0.19 | 0.37 | 3.08 | 34.70 | 1.90 | 0.73 | 66.76 |
| Block3 | 33.65 | 0.40 | 0.17 | 13.63 | 73.39 | 0.87 | 0.65 | 64.98 |
| Block4 | 43.65 | 0.21 | 0.26 | 9.66 | 118.90 | 0.61 | 0.94 | 73.77 |
| Block5 | 53.86 | 0.16 | 0.26 | 9.17 | 159.91 | 0.51 | 0.96 | 84.05 |
| Block6 | 131.84 | 0.53 | 0.19 | **70.54** | 153.54 | 0.49 | 0.94 | 76.64 |
| Block7 | 26.08 | 0.18 | 0.17 | 5.05 | 286.83 | 1.21 | 0.93 | **348.51** |

| | Img64uncond | | | | Img64cond | | | |
|---|---|---|---|---|---|---|---|---|
| | $||\Sigma^{-1}X||$ | $||W_s||$ | $||W_u||$ | CRM | $||\Sigma^{-1}X||$ | $||W_s||$ | $||W_u||$ | CRM |
| Block0 | 53.48 | 0.41 | 0.10 | **22.29** | 82.72 | 1.70 | 0.42 | **141.22** |
| Block1 | 8.00 | 0.11 | 0.12 | 1.06 | 14.26 | 0.59 | 0.78 | 9.25 |
| Block2 | 8.55 | 0.08 | 0.24 | 1.01 | 4.53 | 0.16 | 0.59 | 1.36 |
| Block3 | 13.79 | 0.09 | 0.12 | 1.48 | 7.61 | 0.16 | 0.53 | 1.82 |
| Block4 | 8.36 | 0.06 | 0.25 | 0.77 | 25.09 | 0.29 | 0.33 | 7.68 |
| Block5 | 8.06 | 0.05 | 0.13 | 0.58 | 27.44 | 0.17 | 0.38 | 5.08 |
| Block6 | 40.50 | 0.36 | 0.13 | **14.78** | 12.26 | 0.21 | 0.45 | 3.08 |
| Block7 | 13.97 | 0.12 | .26 | 1.95 | 56.91 | 0.34 | 0.28 | **19.81** |

## D    IGM AND CRM WITH DIFFERENT NORM

We also computed the IGM and CRM under both the Frobenius norm and the 1-norm, shown in Table 7a 7b. The numerical values differ among the Frobenius norm (F), 1-norm, and spectral norm (Table 1 2), while the relative ranks of each block remain entirely consistent. Notably, the spectral norm exhibits a more dispersed distribution in its measurements.

Table 7: IGM and CRM for four models in both F-Norm and 1-Norm

| Models | Img128cond | | Img64cond | | Img64uncond | | AFHQ | |
|---|---|---|---|---|---|---|---|---|
| F-norm | $Z$ | $Z_0$ | $Z$ | $Z_0$ | $Z$ | $Z_0$ | $Z$ | $Z_0$ |
| Block0 | 12.40 | 14.61 | $4.2\times1e5$ | 10.13 | 1416.83 | 8.29 | 21.38 | 23.24 |
| Block1 | 14.83 | 3.92 | 7.95 | 8.66 | 10.67 | 7.08 | 12.79 | 11.23 |
| Block2 | 3.56 | 6.50 | 6.55 | 7.66 | 9.39 | 7.83 | 13.12 | 11.69 |
| Block3 | 10.27 | 13.94 | 5.00 | 7.42 | 6.47 | 5.30 | 35.07 | 44.06 |
| Block4 | 9.37 | 29.58 | 10.63 | 13.86 | 4.02 | 10.19 | 16.39 | 53.62 |
| Block5 | 14.88 | 27.92 | 5.39 | 26.29 | 3.86 | 7.82 | 30.00 | 54.26 |
| Block6 | 53.36 | 41.41 | 4.54 | 19.83 | 33.96 | 7.74 | 14.99 | 60.40 |
| Block7 | 10.27 | 39.87 | 24.04 | 14.17 | 24.98 | 35.99 | 134.16 | 145.32 |
| 1-Norm | $Z$ | $Z_0$ | $Z$ | $Z_0$ | $Z$ | $Z_0$ | $Z$ | $Z_0$ |
| Block0 | 137.76 | 198.3 | $3.5\times1e6$ | 39.77 | 14915.71 | 156.51 | 94.23 | 148.39 |
| Block1 | 167.96 | 53.21 | 39.2 | 49.39 | 204.01 | 116.13 | 71.84 | 113.24 |
| Block2 | 88.66 | 99.38 | 35.58 | 45.56 | 143.53 | 98.95 | 204.35 | 69.05 |
| Block3 | 106.1 | 161.9 | 49.86 | 42.7 | 188.01 | 99.36 | 302.48 | 277.35 |
| Block4 | 97.83 | 551.18 | 95.01 | 156.55 | 61.84 | 169.69 | 158.54 | 1024.6 |
| Block5 | 151.26 | 357.02 | 48.15 | 173.57 | 68.53 | 112.22 | 271.19 | 429.51 |
| Block6 | 634.4 | 454.57 | 38.66 | 228.86 | 1028.43 | 174.63 | 99.23 | 766.3 |
| Block7 | 118.41 | 449.14 | 236.01 | 77.94 | 524.11 | 633.18 | 1698.14 | 1370.99 |

(a) IGM with Frobenius Norm and 1-Norm

| Models | Img128cond | | Img64cond | | Img64uncond | | AFHQ | |
|---|---|---|---|---|---|---|---|---|
| F-Norm | CRM | Percent | CRM | Percent | CRM | Percent | CRM | Percent |
| Block0 | 9.35 | 5.50 | 150.29 | **63.42** | 19.23 | **35.22** | 60.31 | 5.10 |
| Block1 | 10.89 | 6.41 | 14.62 | 6.17 | 2.12 | 3.88 | 74.53 | 6.30 |
| Block2 | 5.72 | 3.37 | 3.66 | 1.54 | 2.25 | 4.13 | 103.02 | 8.72 |
| Block3 | 17.12 | 10.08 | 4.59 | 1.93 | 3.12 | 5.72 | 82.16 | 6.95 |
| Block4 | 16.50 | 9.71 | 13.09 | 5.52 | 1.68 | 3.08 | 104.05 | 8.80 |
| Block5 | 19.34 | 11.38 | 10.82 | 4.57 | 1.43 | 2.62 | 133.83 | 11.33 |
| Block6 | 83.75 | **49.30** | 6.59 | 2.78 | 21.99 | **40.27** | 149.97 | 12.69 |
| Block7 | 7.17 | 4.22 | 33.25 | **14.03** | 2.75 | 5.04 | 473.30 | **40.07** |
| 1-Norm | CRM | Percent | CRM | Percent | CRM | Percent | CRM | Percent |
| Block0 | 99.98 | 3.39 | 799.34 | **45.24** | 219.04 | **25.17** | 1320.45 | 4.56 |
| Block1 | 194.08 | 6.59 | 274.96 | 15.56 | 46.72 | 5.37 | 948.31 | 3.28 |
| Block2 | 84.62 | 2.87 | 15.43 | 0.87 | 25.37 | 2.91 | 1085.57 | 3.75 |
| Block3 | 666.29 | 22.64 | 33.05 | 1.87 | 64.74 | 7.44 | 2364.8 | 8.18 |
| Block4 | 160.53 | 5.45 | 97.42 | 5.51 | 15.79 | 1.81 | 1799.76 | 6.22 |
| Block5 | 345.13 | 11.72 | 144.68 | 8.18 | 23.56 | 2.70 | 4486.86 | 15.52 |
| Block6 | 1253.95 | **42.61** | 52.84 | 2.99 | 341.51 | **39.25** | 2396.26 | 8.28 |
| Block7 | 137.96 | 4.68 | 349.06 | **19.75** | 133.23 | 15.31 | 14507.4 | **50.18** |

(b) CRM with Frobenius Norm and 1-Norm

# E ROBUSTNESS FOR IGM AND CRM

To further verify the robustness of these metrics across categories, we computed CRMs of Img128cond model with 1000 diverse categories in ImageNet. We then calculated:

The percentage of CRMs and its std across the 8 TarFlow blocks and the corresponding ranking of CRMs and its std, we listed the results in table 8.

| CRM | Mixed | | Average over 1000 categories | |
|---|---|---|---|---|
| | Rank | Percent | Rank | Percent |
| Block0 | 3 | 5.22 | $3.63 \pm 1.24$ | $6.50 \pm 2.30$ |
| Block1 | 4 | 5.63 | $5.49 \pm 1.19$ | $8.86 \pm 2.66$ |
| Block2 | 1 | 2.47 | $1.06 \pm 0.24$ | $2.66 \pm 0.25$ |
| Block3 | 7 | 10.93 | $6.72 \pm 0.54$ | $11.01 \pm 0.79$ |
| Block4 | 6 | 7.74 | $5.04 \pm 0.86$ | $7.41 \pm 0.52$ |
| Block5 | 5 | 7.35 | $3.93 \pm 0.86$ | $6.94 \pm 0.50$ |
| Block6 | **8** | **56.57** | **$8 \pm 0$** | **$52.73 \pm 3.92$** |
| Block7 | 2 | 4.05 | $2.12 \pm 0.42$ | $3.89 \pm 0.29$ |

Table 8: CRM performance comparison across different blocks on Img128cond datasets

It can be observed that the dominant CRM layer remains unchanged (std=0) across different categories, and the fluctuations are minimal. Based on these experiments, we speculate that CRM-guided behavior has minimal impact on input image categories, thus using values computed from one class will almost not affect performance on another.

We then calculated IGMs for 8 blocks in Img128cond model with 1000 diverse categories in ImageNet. We recorded:

The means and stds of IGMs by initializing with $Z, Z_0$ and the total counts of $\mathrm{CRM}(Z) < \mathrm{CRM}(Z_0)$ in all 1000 categories.

| IGM | Mixed | | Average over 1000 categories | | Count |
|---|---|---|---|---|---|
| | Z | $Z_0$ | Z | $Z_0$ | |
| Block0 | 12.06 | 14.15 | $16.16 \pm 5.87$ | $20.21 \pm 8.29$ | 958 |
| Block1 | 14.19 | 3.55 | $17.18 \pm 5.93$ | $6.85 \pm 2.92$ | 17 |
| Block2 | 3.35 | 5.66 | $4.30 \pm 1.41$ | $15.25 \pm 7.90$ | 996 |
| Block3 | 10.33 | 14.23 | $11.86 \pm 3.10$ | $15.17 \pm 3.76$ | 989 |
| Block4 | 9.04 | 29.26 | $9.42 \pm 1.51$ | $29.78 \pm 4.16$ | 1000 |
| Block5 | 14.78 | 26.89 | $15.30 \pm 3.32$ | $28.26 \pm 5.94$ | 1000 |
| Block6 | 53.42 | 42.03 | $53.10 \pm 4.32$ | $42.69 \pm 7.21$ | 12 |
| Block7 | 11.00 | 39.67 | $19.71 \pm 5.91$ | $43.66 \pm 4.77$ | 1000 |

Table 9: IGM performance comparison across different blocks

As shown in table 9, the relative size of IGM between $Z, Z_0$ -initialization for each block—computed across all 1000 categories—exhibit stability, with count of $\mathrm{I}(Z < Z_0)$ clustering tightly around 0 and 1000. As discussed in the paper, since GS-Jacobi iteration is guaranteed to converge within finite steps, the influence of input categories on IGM is negligible.

# F  GS-JACOBI SAMPLING

The complete algorithm of GS-Jacobi sampling is as follows:

---

**Algorithm 1** Guass-Seidel-Jacobi Sampling

---

**Input:** Well trained TarFlow model containing $L$ blocks $\{\text{Block}^{\langle l \rangle} := \Sigma^{\langle l \rangle}, \mu^{\langle l \rangle}\}_{l=1}^L$, a batch of training samples $X$, a batch of noise $Z$, other hyperparameters.
**Output:** Generated images of the same size as $Z$.
**Preprocessing:**
  1: Patchify $X$ into size $(B, T, C)$
  2: **for** Block$^{\langle l \rangle}$, $l$ in $1 : L$ **do**
  3:     Calculate IGM$^{\langle l \rangle}$ with equation (4)
  4:     Calculate CRM$^{\langle l \rangle}$ with equation (5)
  5: **end for**
  6: Record the initial guessing mode of $l$-th block to $Z^{\langle l \rangle}$ or $Z_0^{\langle l \rangle}$ according to IGM$^{\langle l \rangle}$
  7: Determine the GS modules numbers $\{G_l\}_{l=1}^L$ and Jacobi times $\{J_l\}_{l=1}^L$ for each blocks according to CRM$^{\langle l \rangle}$, with $J_L \leq T//G_l$
**Sampling:**
  1: Patchify $Z$ into size $(B, T, C)$
  2: Set $Z^{\langle 1 \rangle} = Z$, ebound $= 10^{-8}$
  3: **for** Block$^{\langle l \rangle}$, $l$ in $1 : L$ **do**
  4:     **for** $g$ in $1 : G_l$ **do**
  5:         set $k = 0, e = 1000$
  6:         **while** $k < J_l$ and $e >$ ebound **do**
  7:             $X_g^{\langle l \rangle (k+1)} = \Sigma_g(X_{:g}^{\langle l \rangle (k)}) Z_g^{\langle l \rangle} + \mu_g(X_{:g}^{\langle l \rangle (k)})$
  8:             $e = \|X_g^{\langle l \rangle (k+1)} - X_g^{\langle l \rangle (k)}\| / (B \times T \times C)$
  9:             $k = k + 1$
 10:         **end while**
 11:         $Z^{\langle l+1 \rangle} = X^{\langle l \rangle (k+1)}$
 12:     **end for**
 13: **end for**
 14: **return** Unpatchified $Z^{\langle L+1 \rangle}$

---

An intuition Figure is:

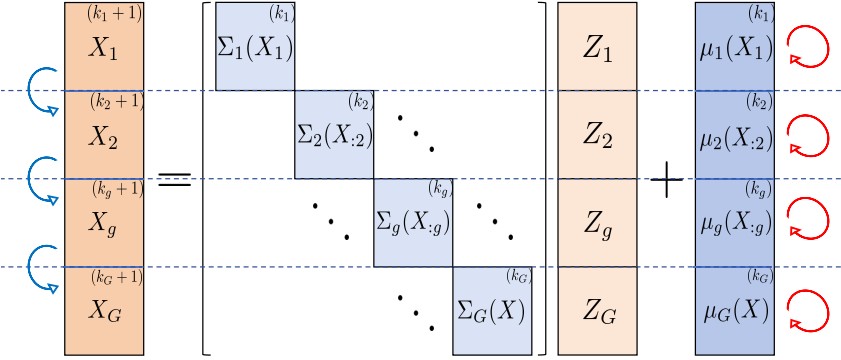

Figure 6: Intuition diagram of Gauss-Seidel-Jacobi sampling in single block. The horizontal long dashed line segment (3) into $G$ subgroups. The red rotating arrow denote the in-group Jacobi iteration with $k_g \leq \text{card}(\mathcal{G}_g)$ times. Then, the solution closed to $X_g^*$ will be delivered to next subgroup serially, this is the Gauss-Seidel part which denoted by blue rotating arrow.

# G   VISUAL COMPARISON OF DIFFERENT METHODS

We used the original "For" loop and GS-Jacobi strategy with FID relative error within 1% for each model under the same guidance and denoise, and visualized the sampling results in Figure 7.

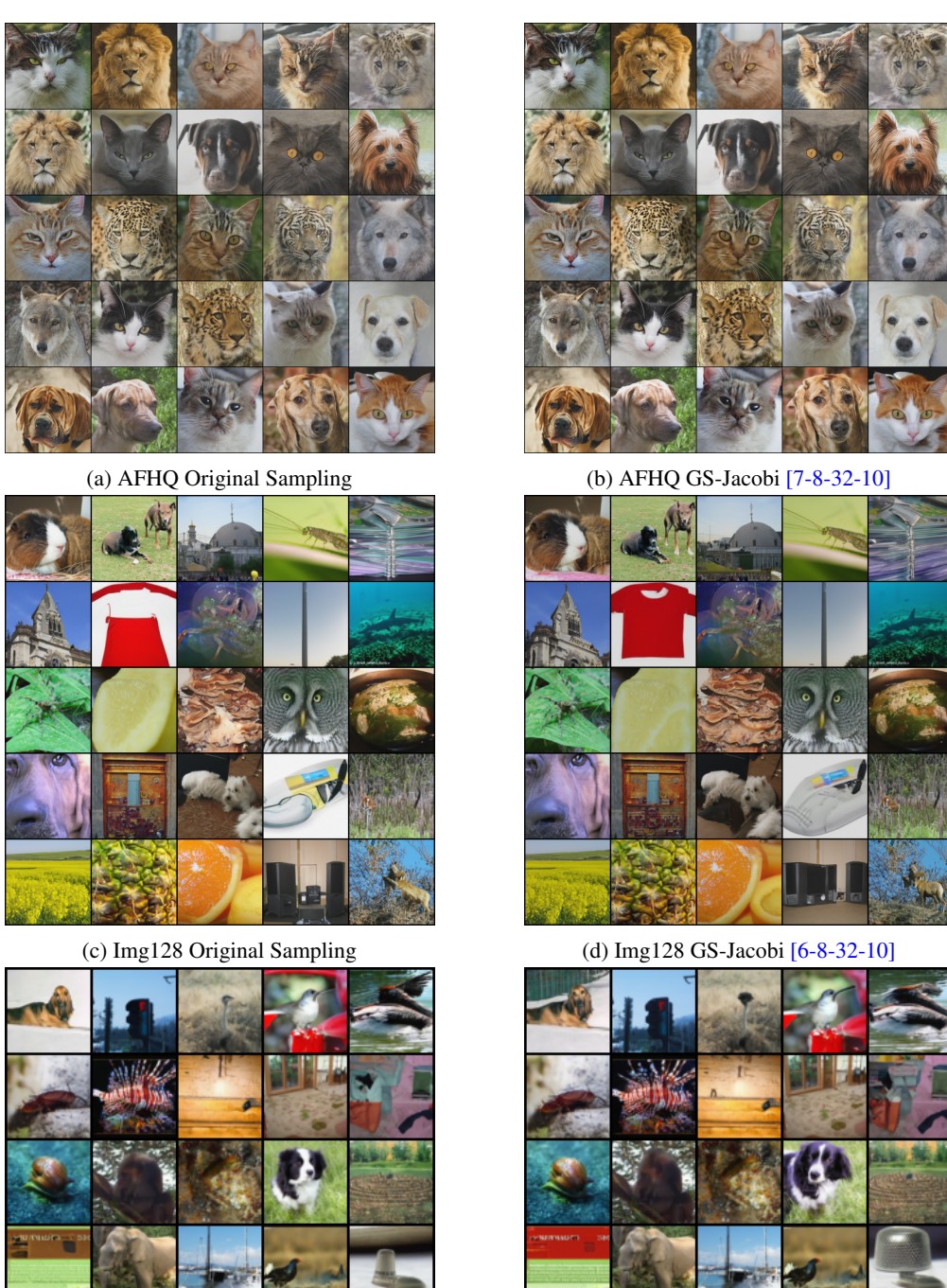

(a) AFHQ Original Sampling

(b) AFHQ GS-Jacobi [7-8-32-10]

(c) Img128 Original Sampling

(d) Img128 GS-Jacobi [6-8-32-10]

(e) Img64 Original Sampling

(f) Img64 GS-Jacobi [0/7-16/8-10/13-6]

Figure 7: Visual Comparison of Different Methods

