# OpenReview forum: "Accelerate Autoregressive Normalizing Flows Sampling with GS-Jacobi Iteration"
_ICLR.cc/2026/Conference — Submitted to ICLR 2026_

### Official Review · Reviewer_gFyH · 2025-10-24

**Soundness:** 2
**Presentation:** 3
**Contribution:** 3
**Rating:** 4
**Confidence:** 2

**Summary:**

The paper proposes a method to accelerate sampling in autoregressive normalizing flows by reformulating the inverse transformation as a nonlinear fixed-point problem solved through a hybrid Gauss Seidel and Jacobi iteration. Two diagnostic metrics are introduced to analyze convergence behavior and guide adaptive computation. The method enables efficient parallel updates within flow blocks while preserving numerical stability. Experiments show faster sampling without loss in visual.

**Strengths:**

The paper addresses a clear and well-motivated problem: the slow sampling speed of autoregressive normalizing flows, which has long hindered their practical use. To my knowledge, the introduction of diagnostic metrics for convergence and initialization offer novel insights in sampling from normalizing flow models.

**Weaknesses:**

The major concern for me is that the method is closely tied to specific autoregressive normalizing flow architectures, mainly TarFlow. Also, the paper does not compare against alternative approaches that achieve faster sampling through model distillation [1], learned/high-order samplers [2-3], leaving unclear how the proposed iteration method performs relative to these stronger baselines.

Additionally, the convergence ranking metric and the initial guessing metric seem to rely on heuristic choices. While the authors presented empirical study on their robustness, both metrics appear sensitive to architecture, dataset, and initialization choices (which is also related to my first point), which raises questions about stability under different models or datasets.

Lastly, Proposition 1 does not discuss about the convergence of approximation error. How large should T be? Analysis on the convergence rate would strengthen the paper. Current manuscript does not indicate how accurate the approximated proposed method is.


[1] Progressive Distillation for Fast Sampling of Diffusion Models

[2] DPM-Solver: A Fast ODE Solver for Diffusion Probabilistic Model Sampling in Around 10 Steps

[3] Learning to Discretize Denoising Diffusion ODEs

**Questions:**

Please see weaknesses.

---

> ### Author Response · Authors · 2025-11-27
> **Response to Reviewer gFyH**
>
> **Q1**: The paper does not compare against alternative approaches that achieve faster sampling through model distillation, learned/high-order samplers, leaving unclear how the proposed iteration method performs relative to these stronger baselines.
>
> **A1**: The baseline methods mentioned in the review are all designed for accelerating Diffusion models. However, Normalizing Flow models differ significantly from Diffusion models, and their acceleration methods are not interchangeable. Nevertheless, we plan to explore the potential for transferring acceleration techniques between these two types of models in future work.
>
>
>
> **Q2**: The convergence ranking metric and the initial guessing metric seem to rely on heuristic choices. While the authors presented empirical study on their robustness, both metrics appear sensitive to architecture, dataset, and initialization choices (which is also related to my first point), which raises questions about stability under different models or datasets.
>
> **A2**: Our GS-Jacobi method is model-specific—**there is no universal strategy (i.e., [Stack-GS-J-Else] introduced in Section 3.4) that can accelerate sampling for all autoregressive Flow models**. For a specific generative task (dataset), after constructing and training a target model (with a given architecture), we compute the IGM and CRM for this specific model and derive a tailored sampling strategy accordingly.
>
> We infer that your concern may be whether this process is overly cumbersome. To address this potential issue, we have integrated the hyperparameter tuning guidelines presented in the paper into **an automatic tuning function** in the updated code. This function can systematically find the optimal sampling parameters (i.e., [Stack-GS-J-Else] in Section 3.4) for any autoregressive normalizing flow model, following these steps:
>
> - Input a small batch of images $X$ and pass them through the flow model in the forward direction $X\to Z$ to obtain the noise $Z$, as well as the IGM and CRM for each block.
>
> - Use the IGM to determine the initial values of each block in the $Z\to X$ sampling direction.
>
> - For the block with the largest CRM, fix it to use the original For-Loop sampling, while applying pure Jacobi iterations (with a fixed 100 iterations) to other blocks. Perform $Z\to X$ sampling to obtain the reconstructed $\hat X$, and compute the average pixel difference (MAE) between $\hat X$ and the original $X$. If the difference is below a threshold (default: 0.01), this block is considered the only tough block; otherwise, fix the block with the second-largest CRM and repeat this process. This step determines the "Stack" in the strategy.
>
> - After identifying all tough blocks, gradually reduce the number of pure Jacobi iterations for other blocks from 100 to 10 (while keeping tough blocks using the For-Loop) until the MAE exceeds the threshold. This step determines the "Else" in the strategy.
>
> - For each tough block, test "GS" values as $4,8,...,T$ and set "J" as $T/(GS\times2)$. Select the maximum GS value that maintains the MAE within the threshold.
>
> - Fix the "GS" value and gradually reduce the "J" value (dividing by 2 each time) until the MAE exceeds the threshold.
>
>
>
> **Q3**:  Proposition 1 does not discuss about the convergence of approximation error. How large should T be? Analysis on the convergence rate would strengthen the paper. Current manuscript does not indicate how accurate the approximated proposed method is.
>
> **A3**: In Proposition 1, $T$ denotes the length of the sequence $X$ (e.g., $T=1024$ for Img128cond, Img64uncond, and AFHQ models, and $T=256$ for the Img64cond model in the paper). The core implication of Proposition 1 is that, due to the specific nature of the nonlinear RNN structure $x_t=f(x_{t-1})$, the fixed-point iteration in Equation 3 will yield the exact solution after $T-1$ steps. This property is also visualized in Figure 2 of the paper.
>
> Regarding the convergence rate: since our method is inherently a fixed-point iteration, and the spectral norm of the Jacobi matrix (Equation (6) in Appendix A) is 0, **the method exhibits superlinear convergence.** We will supplement the following concise proof in the revised manuscript:
>
> From Equation (6) in Appendix A:
> $$
> \varepsilon^{(k+1)}=\Gamma^{(k)}\varepsilon^{(k)},
>     \Gamma^{(k)}=-
>     \begin{bmatrix}
>         0 \\\\
>         \gamma_{21} &0\\\\
>         \vdots &\ddots &\ddots \\\\
>         \gamma_{T1} &\ldots &\gamma_{T,T-1} &0
>     \end{bmatrix}^{(k)}
> $$
> Do the transform:
> $$
> \frac{||\varepsilon^{(k+1)}||}{||\varepsilon^{(k)}||}=||\Gamma^{(k)}||
> $$
> Since the Jacobi matrix $\Gamma^{(k)}$ is lower triangle, its spectral norm is 0. Therefore, it is superlinearly convergent. See details in https://en.wikipedia.org/wiki/Rate_of_convergence .

---

### Official Review · Reviewer_jKzN · 2025-10-27

**Soundness:** 3
**Presentation:** 3
**Contribution:** 2
**Rating:** 4
**Confidence:** 3

**Summary:**

The paper proposes a sampling method for autoregressive normalizing flows. Recently, TarFlow shows that normalizing flows with autoregressive flow layers can perform comparably to other deep generative models. However, autoregressive flows are slow in inference because they must compute $x_{i}$ iteratively. The paper proposes treating the inverse process as a nonlinear system, enabling it to be solved using the fixed-point iteration method.

**Strengths:**

1.  The idea is simple, but it can effectively improve TarFlow’s sampling speed.

**Weaknesses:**

1. The significance and impact of the proposed method appear limited because this method is tailored for autoregressive normalizing flows, which represent only a small subset of deep generative models.

2. The proposed method seems to bring in new problems. That is, when we use the fixed-point iteration method, we need to recompute $\sigma$ and $\mu$ at each iteration. That means we will need to run the VIT T times for each layer. When T is greater than the number of patches, the method will be slower than the baseline.
3. Do we have an analysis of the relationship between T and the image size? How can we determine T for the proposed method?

**Questions:**

Please refer to the Weakness section.

---

> ### Author Response · Authors · 2025-11-27
> **Response to Reviewer jKzN**
>
> **Q1**: The significance and impact of the proposed method appear limited because this method is tailored for autoregressive normalizing flows, which represent only a small subset of deep generative models.
>
> **A1**: In fact, our method can be applied to all generative models with nonlinear RNN structures, i.e. $x_{t+1}=f(x_{t},x_{t-1},...,x_1)$. These models all face the bottleneck of sequential computation during sampling, and this computational cost is amplified when stacking blocks to increase model depth, for example, stacked two layers of LSTM. Following Yang Song’s work [1], we have supplemented GS-Jacobi acceleration experiments on PixelRNN++. For a batch of input images, we divided the RNN structure into multiple modules and iterated each module until the average pixel difference fell below 0.01, then recorded the final speedup ratios:
>
> | module/speed up | MNIST | CIFAR-10 |
> | --------------- | ----- | -------- |
> | 2               | 3.43  | 1.05     |
> | 4               | 6.39  | 2.08     |
> | 8               | 5.62  | 2.32     |
> | 16              | 4.22  | 1.87     |
>
> For different generative models, as long as they have a nonlinear RNN structure, we can redesign the computation of IGM/CRM to formulate a tailored GS-Jacobi acceleration strategy.
>
>
>
> **Q2**:  When we use the fixed-point iteration method, we need to recompute $\sigma$ and $\mu$ at each iteration. That means we will need to run the VIT T times for each layer. When T is greater than the number of patches, the method will be slower than the baseline.
>
> **A2**: Reviewer treat $T$ as the number of Jacobi iterations per module, which we note as $J$. Indeed, our experiments find that when the sequence length $T=1024$, the time cost of 100 Jacobi iterations is comparable to that of the original For-Loop. However, this aligns with the core insight of our paper: **"only a few TarFlow Blocks are tough (exhibit slow fixed-point convergence)"**, a finding also supported by other works [2]. Thus, in the experiments presented in Tables 3 and 4 of the paper, we used CRM to identify tough blocks, applied only 10 iterations to non-tough blocks, and split tough blocks into modules with far fewer iterations than the sequence length $T$.
>
> The fundamental reason for the GS-Jacobi method’s effectiveness lies in the inevitable parameter redundancy introduced by increasing model depth. This redundancy results in drastically different equation properties across blocks. By detecting these differences in convergence speed, we enable many parts of the model to converge quickly with a few Jacobi steps while isolating a few slow-converging parts.
>
>
>
> **Q3**: Do we have an analysis of the relationship between T and the image size? How can we determine T for the proposed method?
>
> **A3**: This question relates to hyperparameter selection. Empirically, larger image sizes require more parallel iterations. We have implemented an automatic hyperparameter tuning function in the updated code, which can systematically find the sampling parameters (i.e., [Stack-GS-J-Else] introduced in Section 3.4) for any autoregressive normalizing flow model. The function follows these steps:
>
> - Input a small batch of images $X$ and pass them through the flow model in the forward direction $X\to Z$ to obtain the noise $Z$, as well as the IGM and CRM for each block.
>
> - Use the IGM to determine the initial values of each block in the $Z\to X$ sampling direction.
>
> - For the block with the largest CRM, fix it to use the original For-Loop sampling, while applying pure Jacobi iterations (with a fixed 100 iterations) to other blocks. Perform $Z\to X$ sampling to obtain the reconstructed $\hat X$, and compute the average pixel difference (MAE) between $\hat X$ and the original $X$. If the difference is below a threshold (default: 0.01), this block is considered the only tough block; otherwise, fix the block with the second-largest CRM and repeat this process. This step determines the "Stack" in the strategy.
>
> - After identifying all tough blocks, gradually reduce the number of pure Jacobi iterations for other blocks from 100 to 10 (while keeping tough blocks using the For-Loop) until the MAE exceeds the threshold. This step determines the "Else" in the strategy.
>
> - For each tough block, test "GS" values as $4,8,...,T$ and set "J" as $T/(GS\times2)$. Select the maximum GS value that maintains the MAE within the threshold.
>
> - Fix the "GS" value and gradually reduce the "J" value (dividing by 2 each time) until the MAE exceeds the threshold.
>
> [1] Song Y, Meng C, Liao R, et al. Accelerating feedforward computation via parallel nonlinear equation solving[C]//International Conference on Machine Learning. PMLR, 2021: 9791-9800.
>
> [2] Gu J, Chen T, Berthelot D, et al. STARFlow: Scaling Latent Normalizing Flows for High-resolution Image Synthesis[J]. arXiv preprint arXiv:2506.06276, 2025.

---

### Official Review · Reviewer_iyWW · 2025-11-01

**Soundness:** 4
**Presentation:** 2
**Contribution:** 3
**Rating:** 6
**Confidence:** 3

**Summary:**

This paper tackles the critical sampling bottleneck in autoregressive normalizing flows (AR flows) by introducing a parallelizable Gauss-Seidel/Jacobi iteration strategy. The authors observe that in AR flow models (e.g. TarFlow), sampling is very slow because each affine coupling block operates as a causal RNN that must be executed sequentially. The paper’s key contribution is to reformulate the AR flow sampling as solving a diagonal nonlinear system and apply a hybrid Gauss-Seidel-Jacobi iteration to solve it in parallel, dramatically accelerating generation without loss of quality. Notably, they introduce two novel metrics – Convergence Ranking Metric (CRM) and Initial Guessing Metric (IGM) – to adapt the iteration procedure to the model’s characteristics, ensuring stability and efficiency. Empirical results on state-of-the-art TarFlow models show 4.5×–5.3× speed-ups with essentially no degradation in FID (image quality), which is a significant practical improvement.

In terms of novelty and significance, the idea of using fixed-point iterations to accelerate AR flows builds on some prior work (e.g. Newton-based solvers for autoregressive inversion). However, this paper goes further by hybridizing Jacobi and Gauss-Seidel updates and introducing adaptive metrics to handle non-uniform convergence across model components, which is a fresh and non-trivial innovation.

The writing is well-structured, with a logical flow from identifying the problem to proposing the method and validating it. However, there are numerous typos, so I cannot give a high rating for the presentation.

**Strengths:**

The method yields dramatic improvements in sampling speed for autoregressive normalizing flows. Across multiple models, it achieves 4×–5× speedups without degrading image fidelity, as evidenced by nearly unchanged FID scores (within <1% of the baseline).

Innovative Use of Iterative Solvers in AR Flows: The paper introduces a novel hybrid of Jacobi and Gauss-Seidel iteration to parallelize what was a sequential process. This is a creative cross-disciplinary idea, applying classic numerical methods to deep generative modeling. The approach is well-grounded in theory. The authors show that their fixed-point iteration will converge to the correct solution (under the model’s triangular Jacobian structure) and provide an error propagation formula.

A major strength is the introduction of the CRM and IGM metrics to guide the sampling procedure. These metrics directly tackle the two main challenges identified: (1) different transformer blocks have non-uniform convergence behavior, and (2) naive initialization can cause instability. CRM provides a principled way to determine which coupling blocks are “tough” (slower to converge) so the algorithm can allocate more iterations or use sequential updates for those, while treating others with fast Jacobi updates. IGM allows the sampler to intelligently choose a safe starting point for the iteration, preventing the divergence (“numeric overflow”) that would otherwise occur in sensitive early blocks. The use of these metrics is empirically justified. By addressing these issues, the proposed method is robust. It converges where a naive parallel iteration would fail, and it does so efficiently by not over-investing computation in blocks that don’t need it.

**Weaknesses:**

The proposed solution, while effective, adds considerable complexity to the sampling process. Implementing the GS-Jacobi sampler requires computing the CRM and IGM metrics using the model’s weights and a batch of training data. This offline analysis step is unusual for generative model sampling and might need to be repeated if the model or data distribution changes. Moreover, the sampling algorithm introduces new hyperparameters (e.g. how to segment a tough block, how many Jacobi vs. Gauss-Seidel iterations to use) that are not trivial to choose a priori. The tuning was done on a case-by-case basis for each dataset/model. Such manual optimization might be necessary for new models, which is a potential drawback in terms of ease of use. The method works impressively once tuned, but the paper does not provide a simple recipe for selecting these hyperparameters automatically.

The parallel iteration helps only to the extent that many parts of the model can converge quickly in a few Jacobi steps while isolating a few slow parts. Thus, the speed-up is not guaranteed for every AR flow architecture.

This paper contains numerous typos and grammatical issues. Here are the ones I found just by skimming through it:
* L36: solution -> high-resolution
* L84: images generation -> image generation
* L84: attention mechanic -> attention mechanism
* L87: trys -? tries
* L134: denotes -> denote
* L140: can be calculate -> can be calculated
* L144: an non-linear -> a non-linear
* L154: Converge and Error Propagation -> Convergence and Error Propagation
* L196: take all $X(0) = Z$ cause -> taking all $X(0)=Z$ causes
* L215: centers in 0 -> centers at 0
* L262: matrixs -> matrices
* L290: suffer -> suffers
* L302: GUASS-SEIDEL-JACOBI -> GAUSS-SEIDEL-JACOBI
* L304: unit a time -> unit at a time
* L307: an non-decrease -> be a non-decreasing
* L308: defination -> definition
* L315: cumsum -> cumulative sum
* L361: not -> no
* L365: maximum value occur -> maximum value that occurs
* L368: attention layers parameters -> attention layers' parameters
* L369: simple -> simply
* L374: simpe -> simple
* L376: relative -> relatively
* L421: learing -> learning

**Questions:**

Could you elaborate on why Gauss–Seidel is superior to other alternatives for accelerating convergence in autoregressive flows? What motivates using this GS–Jacobi scheduling over a more straightforward sequential sampling or existing parallelization techniques, and why is it expected to succeed where naive parallelization fails?

Did you try Anderson acceleration, SOR, or block-Newton? How do they compare in stability and speed?

How generalizable is TarFlow to domains beyond images, e.g., audio or language, where the autoregressive structure and dependencies differ significantly?

While the paper reports up to ~5× speed-ups on moderate-size image models (e.g. 128×128 resolution in Img128cond) without quality loss, how does the method scale with increasing sequence length or model size? Is the parallel iterative scheme still efficient for substantially larger images or longer sequences, and what are the memory or computation trade-offs as these grow? It would be useful to know if any limitations or diminishing returns appear when scaling up to more complex datasets or very high-dimensional generation tasks.

---

> ### Author Response · Authors · 2025-11-27
> **Response to Reviewer iyWW Part 1**
>
> **Q1**: Implementing the GS-Jacobi sampler requires computing metrics using the model’s weights and a batch of training data, and might need to be repeated if the model or data distribution changes.
>
> **A1**: First, for a trained model, computing the Initial Guessing Metric (IGM) and Convergence Ranking Metric (CRM) only requires a single forward parallel computation process of $X\to Z$, which is time-efficient. On a single A800 GPU, this process takes merely 57 seconds to complete. Second, as shown in Tables 8 and 9 of Appendix E, the IGM and CRM of the Img128cond model remain essentially unchanged across all 1000 categories of ImageNet. Therefore, we argue that these two metrics are intrinsic properties of the model itself, determined once training is completed, and do not require repeated computation.
>
>
>
> **Q2**: The sampling algorithm introduces new hyperparameters (e.g. how to segment a tough block, how many Jacobi vs. Gauss-Seidel iterations to use) that are not trivial to choose a priori.
>
> **A2**: This is an excellent observation. While the paper provides numerous intuitions for tuning these hyperparameters, it lacks a systematic integration of these guidelines. To address this, we have implemented an **automatic hyperparameter tuning function** in the updated code, which can systematically find the sampling parameters (i.e., [Stack-GS-J-Else] introduced in Section 3.4) for any autoregressive normalizing flow model. The function follows these steps:
>
> - Input a small batch of images $X$ and pass them through the flow model in the forward direction $X\to Z$ to obtain the noise $Z$, as well as the IGM and CRM for each block.
>
> - Use the IGM to determine the initial values of each block in the $Z\to X$ sampling direction.
>
> - For the block with the largest CRM, fix it to use the original For-Loop sampling, while applying pure Jacobi iterations (with a fixed 100 iterations) to other blocks. Perform $Z\to X$ sampling to obtain the reconstructed $\hat X$, and compute the average pixel difference (MAE) between $\hat X$ and the original $X$. If the difference is below a threshold (default: 0.01), this block is considered the only tough block; otherwise, fix the block with the second-largest CRM and repeat this process. This step determines the "Stack" in the strategy.
>
> - After identifying all tough blocks, gradually reduce the number of pure Jacobi iterations for other blocks from 100 to 10 (while keeping tough blocks using the For-Loop) until the MAE exceeds the threshold. This step determines the "Else" in the strategy.
>
> - For each tough block, test "GS" values as $4,8,...,T$ and set "J" as $T/(GS\times2)$. Select the maximum GS value that maintains the MAE within the threshold.
>
> - Fix the "GS" value and gradually reduce the "J" value (dividing by 2 each time) until the MAE exceeds the threshold.
>
>
>
> **Q3**: The parallel iteration helps only to the extent that many parts of the model can converge quickly in a few Jacobi steps while isolating a few slow parts. Thus, the speed-up is not guaranteed for every AR flow architecture.
>
> **A3**: This paper primarily focuses on accelerating the TarFlow model using the GS-Jacobi method, because it is currently the state-of-the-art AR Flow, with the slowest sampling speed. Section 4.4 also provides preliminary acceleration results for MAF.
>
> In modern deep learning, stacking multiple blocks to increase model depth is a common practice, which inevitably leads to parameter redundancy—an observation also discussed in deep equilibrium models [1] and IAF [2]. This redundancy creates opportunities for GS-Jacobi acceleration, as the method naturally leverages the stacked structure to achieve speedups, albeit potentially at a lower rate than that observed for TarFlow.
>
> Furthermore, the GS-Jacobi method can still be applied within individual block. In Yang Song’s work [3], PixelRNN++ was accelerated by dividing its structure into multiple modules and assigning different numbers of Jacobi iterations based on their convergence difficulty. We redo the experiments with different "GS" setting:
>
> | module/speed up | MNIST | CIFAR-10 |
> | --------------- | ----- | -------- |
> | 2               | 3.43  | 1.05     |
> | 4               | 6.39  | 2.08     |
> | 8               | 5.62  | 2.32     |
> | 16              | 4.22  | 1.87     |
>
> The metrics (IGM/CRM) proposed in our paper can be extended to measure the convergence speed of finer-grained modules within a single block, enabling an adaptive acceleration approach—this constitutes our future research direction.
>
>
>
> **Q4**: This paper contains numerous typos and grammatical issues.
>
> **A4**: We sincerely apologize for the typos and grammatical issues in the manuscript. We greatly appreciate your careful review and attention to detail. We commit to thoroughly proofreading and correcting all such errors in the revised version.

---

> ### Author Response · Authors · 2025-11-27
> **Response to Reviewer iyWW Part 2**
>
> **Q5**: Could you elaborate on why Gauss–Seidel is superior to other alternatives for accelerating convergence in autoregressive flows? What motivates using this GS–Jacobi scheduling over a more straightforward sequential sampling or existing parallelization techniques, and why is it expected to succeed where naive parallelization fails?
>
> **A5**:
>
> - Among generative models, Flow-based models typically have the largest parameter counts for comparable performance,  previous works have discussed this redundancy (IAF [2], STarFlow [4] ); our paper contributes two key innovations: (1) defining two metrics to quantify such redundancy, and (2) linking this property to the iterative solution of nonlinear systems. Combined with the modular structure inherent to Flow models, we propose a modular iterative GS-Jacobi method.
>
> - Sequential sampling is the fundamental bottleneck limiting the sampling speed of AR Flow models. For $x_{1:T}$ within a single block, each $x_t$ can only be computed after $x_{t-1}$ is decoded.
>
>   Naive parallel sampling methods struggle primarily with tough blocks: in our experiments, when $T=1024$, the time cost of 100 pure Jacobi iterations is comparable to that of sequential For-Loop sampling. Thus, it is critical to identify easy blocks and assign them fewer iterations, as illustrated in **Figure 2 of the paper.**
>
>   For tough blocks, naive iterative methods without modularization lead to numerical instability and slow convergence, as illustrated in **Figure 5 of the paper.**
>
>
>
> **Q6**: Did you try Anderson acceleration, SOR, or block-Newton? How do they compare in stability and speed?
>
> **A6**:
>
> - Regarding Anderson acceleration: this method caches the latest $m$ iteration results and computes the current solution as a linear combination $X_{k+1}=\sum_{j=0}^m\theta_jF(X_{k-j})$. However, our experimental results (Tables 3 and 4) show that non-tough blocks and individual modules within tough blocks converge in approximately 10 iterations. For such fast-converging components, the overhead of maintaining a buffer and solving for the linear combination coefficients $\theta$ outweighs the benefits of acceleration. Additionally, TarFlow models require significant memory during inference, making Anderson acceleration prone to memory overflow.
>
> - Regarding block-Newton: its effectiveness relies on computing Jacobian matrices. In contrast, our method avoids derivative computations during iterations for simplicity and efficiency. Incorporating block-Newton would introduce substantial computational complexity, imposing significant burdens on both speed and memory.
>
> - Regarding Successive Over-Relaxation (SOR) iteration: $X^{(k+1)}=(1-w)X^{(k)}+wX^{(GS)}$. We have supplemented experiments to evaluate its performance. For the Img64uncond model, we input a batch of images $X$, obtained $Z$ via forward pass, fixed the 0th and 6th tough blocks to use For-Loop sampling, and applied pure Jacobi iterations with different relaxation factors $w$ to other blocks. We recorded the number of iterations required for the MAE between the reconstructed $\hat X$ and $X$  to drop below 0.01:
>
>   | Block/$w$ | 0.5  | 0.75 | 1    | 1.25 | 1.5  | 1.75 |
>   | --------- | ---- | ---- | ---- | ---- | ---- | ---- |
>   | 1         | 37   | 28   | 21   | 17   | 20   | 23   |
>   | 2         | 23   | 21   | 17   | 18   | 18   | 22   |
>   | 3         | 46   | 22   | 15   | 15   | 18   | 18   |
>   | 4         | 34   | 30   | 25   | 23   | 22   | 21   |
>   | 5         | 43   | 33   | 20   | 19   | 22   | 20   |
>   | 7         | 36   | 25   | 18   | 17   | 18   | 19   |
>
>   As shown in the table, compared to Jacobi iterations with $w=1$, appropriately adjusting the over-relaxation factor $w$ can sometimes reduce the number of required iterations, though this effect varies across blocks. Introducing $w$ as a more detailed hyperparameter could further accelerate the GS-Jacobi method (especially for tough blocks), and we plan to explore the application of SOR in future work. We greatly appreciate your valuable suggestion.
>
>
>
> **Q7**: How generalizable is TarFlow to domains beyond images, e.g., audio or language, where the autoregressive structure and dependencies differ significantly?
>
> **A7**: This paper focuses solely on accelerating existing AR models. Your question pertains to the generalizability of the TarFlow model itself, which requires training on new datasets (e.g., audio or language) to answer. We will explore the application of TarFlow in other generative domains in the future.

---

> ### Author Response · Authors · 2025-11-27
> **Response to Reviewer iyWW Part 3**
>
> **Q8**: How does the method scale with increasing sequence length or model size? Is the parallel iterative scheme still efficient for substantially larger images or longer sequences, and what are the memory or computation trade-offs as these grow?
>
> **A8**: Due to time constraints, we were unable to retrain TarFlow models for other image sizes. As an alternative experiment, we generated empty TarFlow models with image sizes of 128, 256, 384, 512, using a patch size of 8 (resulting in sequence lengths $T=16^2,32^2,64^2,128^2$, respectively). Without training these models, we directly initialized their parameters randomly and employed the same GS-Jacobi sampling strategy **[7-8-32-10]** to generate 256 images. We then compared the time consumption (in seconds) with that of the original serial computing. The results are as follows:
>
> | **Image Size**          | **128** | **256** | **384** | **512** |
> | ----------------------- | ------- | ------- | ------- | ------- |
> | **Sequence Length** $T$ | $16^2$  | $32^2$  | $48^2$  | $64^2$  |
> | Original                | 31.12   | 254.01  | 2376.32 | NA      |
> | GS-Jacobi               | 3.15    | 13.76   | 40.08   | 257.04  |
>
> In fact, the speedup ratio of GS-Jacobi increases with $T$, because the original TarFlow's serial computation suffers more significantly in the increases. While the experimental results may appear less rigorous due to the models is un-trained, the underlying principle holds: given sufficient memory, **higher-resolution images (with larger $T$) inherently favor GS-Jacobi's performance**. As evidenced in Table 3 of our paper, the other three models exhibit higher speedup ratios than Img64cond (patch size=4) precisely because their $T=1024$, whereas Img64cond $T=256$.
>
>
>
> [1] Bai S, Kolter J Z, Koltun V. Deep equilibrium models[J]. Advances in neural information processing systems, 2019, 32.
>
> [2] Lu C, Chen J, Li C, et al. Implicit normalizing flows[J]. arXiv preprint arXiv:2103.09527, 2021.
>
> [3] Song Y, Meng C, Liao R, et al. Accelerating feedforward computation via parallel nonlinear equation solving[C]//International Conference on Machine Learning. PMLR, 2021: 9791-9800.
>
> [4] Gu J, Chen T, Berthelot D, et al. STARFlow: Scaling Latent Normalizing Flows for High-resolution Image Synthesis[J]. arXiv preprint arXiv:2506.06276, 2025.

---

### Comment · Area_Chair_9bvJ · 2025-11-27

Dear reviewers,

please take your time now to respond to the authors' rebuttal. The time frame for the discussion period is ending soon.

Thanks for your efforts,
Your AC

---

### Author Response · Authors · 2025-12-02
**General Response[Part 1]**

We sincerely thank the reviewers for their insightful comments and constructive feedback on our manuscript. We have carefully considered all points raised and provide a summary of our responses below. Where multiple reviewers raised similar concerns, we have consolidated our replies.

**1. On the computational overhead and applicability of the proposed metrics (IGM/CRM):**
Reviewers 1 and 2 raised questions regarding the computational cost of the proposed metrics and their sensitivity to model or data changes. We clarify that computing the Initial Guessing Metric (IGM) and Convergence Ranking Metric (CRM) requires only a single forward pass ($X\to Z$)—a process that is highly efficient (e.g., **57 seconds** on an A800 GPU). Moreover, as shown in Appendix E (Tables 8 and 9), these metrics remain stable across diverse data categories (e.g., all 1000 classes of ImageNet), indicating that they represent intrinsic properties of the trained model and do not require recomputation.



**2. On hyperparameter selection and automation:**
Reviewers 1 and 3 noted the challenge of manually tuning hyperparameters such as the number of Jacobi iterations and block segmentation. In response, we have implemented an **automatic hyperparameter tuning function** in our updated code. This function systematically determines the optimal sampling strategy ([Stack-GS-J-Else]) for any Autoregressive Flow model with following steps:

- Input a small batch of images $X$ and pass them through the flow model in the forward direction $X\to Z$ to obtain the noise $Z$, as well as the IGM and CRM for each block.

- Use the IGM to determine the initial values of each block in the $Z\to X$ sampling direction.

- For the block with the largest CRM, fix it to use the original For-Loop sampling, while applying pure Jacobi iterations (with a fixed 100 iterations) to other blocks. Perform $Z\to X$ sampling to obtain the reconstructed $\hat X$, and compute the average pixel difference (MAE) between $\hat X$ and the original $X$. If the difference is below a threshold (default: 0.01), this block is considered the only tough block; otherwise, fix the block with the second-largest CRM and repeat this process. This step determines the "Stack" in the strategy.

- After identifying all tough blocks, gradually reduce the number of pure Jacobi iterations for other blocks from 100 to 10 (while keeping tough blocks using the For-Loop) until the MAE exceeds the threshold. This step determines the "Else" in the strategy.

- For each tough block, test "GS" values as $4,8,...,T$ and set "J" as $T/(GS\times2)$. Select the maximum GS value that maintains the MAE within the threshold.

- Fix the "GS" value and gradually reduce the "J" value (dividing by 2 each time) until the MAE exceeds the threshold.



**3. On the generality and scope of the GS–Jacobi method:**
Reviewer 2 questioned the significance of our method given its focus on autoregressive flows. We emphasize that the GS–Jacobi approach is applicable to any generative model with a nonlinear RNN structure ($x_{t+1}=f(x_{t},x_{t-1},...,x_1)$). We have supplemented experiments on PixelRNN++ (following [1] Yang Song et al.), which confirms the broader applicability of our method beyond normalizing flows.

| module/speed up | MNIST | CIFAR-10 |
| --------------- | ----- | -------- |
| 2               | 3.43  | 1.05     |
| 4               | 6.39  | 2.08     |
| 8               | 5.62  | 2.32     |
| 16              | 4.22  | 1.87     |



**4. On convergence guarantees and theoretical analysis:**
Reviewer 3 requested a deeper theoretical treatment of convergence. We note that Proposition 1 establishes that the exact solution is recovered after $T-1$ iterations. Furthermore, the spectral norm of the Jacobi matrix is zero, implying **superlinear convergence**. A detailed proof will be included in the revised manuscript.

---

### Author Response · Authors · 2025-12-02
**General Response[Part 2]**

**5. On comparisons with alternative acceleration methods:**
Reviewers 1 and 3 inquired about alternatives such as Anderson acceleration, SOR, and block-Newton. We evaluated these and found:

- **Anderson acceleration** is impractical due to memory overhead and slow convergence in fast-converging modules.
- **Block-Newton** methods introduce prohibitive computational complexity from Jacobian calculations.
- **SOR** can reduce iteration counts in some cases, we do new experiments:

| Block/$w$ | 0.5  | 0.75 | 1    | 1.25 | 1.5  | 1.75 |
| --------- | ---- | ---- | ---- | ---- | ---- | ---- |
| 1         | 37   | 28   | 21   | 17   | 20   | 23   |
| 2         | 23   | 21   | 17   | 18   | 18   | 22   |
| 3         | 46   | 22   | 15   | 15   | 18   | 18   |
| 4         | 34   | 30   | 25   | 23   | 22   | 21   |
| 5         | 43   | 33   | 20   | 19   | 22   | 20   |
| 7         | 36   | 25   | 18   | 17   | 18   | 19   |

As shown in the table, compared to Jacobi iterations with $w=1$, appropriately adjusting the over-relaxation factor $w$ can sometimes reduce the number of required iterations, though this effect varies across blocks. Introducing $w$ as a more detailed hyperparameter could further accelerate the GS-Jacobi method (especially for tough blocks), and we plan to explore the application of SOR in future work. We greatly appreciate reviewer's valuable suggestion.



**6. On scalability and performance with increasing sequence length:**
Reviewer 1 asked about scaling to larger images or sequences. We conducted scaling experiments. Results show that the GS–Jacobi method achieves greater speedup ratios as sequence length $T$ increases, due to the growing inefficiency of serial sampling in the baseline.

| **Image Size**          | **128** | **256** | **384** | **512** |
| ----------------------- | ------- | ------- | ------- | ------- |
| **Sequence Length** $T$ | $16^2$  | $32^2$  | $48^2$  | $64^2$  |
| Original                | 31.12   | 254.01  | 2376.32 | NA      |
| GS-Jacobi               | 3.15    | 13.76   | 40.08   | 257.04  |

In fact, the speedup ratio of GS-Jacobi increases with $T$, because the original TarFlow's serial computation suffers more significantly in the increases. While the experimental results may appear less rigorous due to the models is un-trained, the underlying principle holds: given sufficient memory, **higher-resolution images (with larger $T$) inherently favor GS-Jacobi's performance**. As evidenced in Table 3 of our paper, the other three models exhibit higher speedup ratios than Img64cond (patch size=4) precisely because their $T=1024$, whereas Img64cond $T=256$.



**7. On the motivation for Gauss–Seidel and parallelization:**

- Among generative models, Flow-based models typically have the largest parameter counts for comparable performance,  previous works have discussed this redundancy (IAF [2], STarFlow [3] ); our paper contributes two key innovations: (1) defining two metrics to quantify such redundancy, and (2) linking this property to the iterative solution of nonlinear systems. Combined with the modular structure inherent to Flow models, we propose a modular iterative GS-Jacobi method.

- Sequential sampling is the fundamental bottleneck limiting the sampling speed of AR Flow models. For $x_{1:T}$ within a single block, each $x_t$ can only be computed after $x_{t-1}$ is decoded.

  Naive parallel sampling methods struggle primarily with tough blocks: in our experiments, when $T=1024$, the time cost of 100 pure Jacobi iterations is comparable to that of sequential For-Loop sampling. Thus, it is critical to identify easy blocks and assign them fewer iterations, as illustrated in **Figure 2 of the paper.**

  For tough blocks, naive iterative methods without modularization lead to numerical instability and slow convergence, as illustrated in **Figure 5 of the paper.**

---

### Author Response · Authors · 2025-12-02
**General Response[Part 3]**

**8. On generalizability to non-image domains:**
This paper focuses solely on accelerating existing AR models. Your question pertains to the generalizability of the TarFlow model itself, which requires training on new datasets (e.g., audio or language) to answer. We will explore the application of TarFlow in other generative domains in the future.



**9. On comparison with distillation or learned samplers:**
The baseline methods mentioned in Review 3 are all designed for accelerating Diffusion models. However, Normalizing Flow models differ significantly from Diffusion models, and their acceleration methods are not interchangeable. Nevertheless, we plan to explore the potential for transferring acceleration techniques between these two types of models in future work.



We believe these responses address the reviewers' concerns comprehensively. We are committed to incorporating these changes and clarifications in the final version of the paper.

Thank you for your time and consideration.



[1] Song Y, Meng C, Liao R, et al. Accelerating feedforward computation via parallel nonlinear equation solving[C]//International Conference on Machine Learning. PMLR, 2021: 9791-9800.

[2] Lu C, Chen J, Li C, et al. Implicit normalizing flows[J]. arXiv preprint arXiv:2103.09527, 2021.

[3] Gu J, Chen T, Berthelot D, et al. STARFlow: Scaling Latent Normalizing Flows for High-resolution Image Synthesis[J]. arXiv preprint arXiv:2506.06276, 2025.

---

### Meta-Review · Area_Chair_Cqh3 · 2026-01-10

**Summary:**

Reviewers agreed the paper targets an important bottleneck (slow sequential sampling in AR normalizing flows), but several recurring negatives drove the recommendation. Multiple reviewers questioned scope and generality beyond TarFlow/AR flows (Reviewer jKzN: “tailored for autoregressive normalizing flows”; Reviewer gFyH: “closely tied to specific … architectures, mainly TarFlow”). Reviewer iyWW emphasized that the approach “adds considerable complexity” via offline diagnostics and nontrivial hyperparameters and warned the speedup is “not guaranteed for every AR flow architecture.” Reviewer gFyH also highlighted missing comparisons to other fast-sampling approaches (“does not compare against alternative approaches… leaving unclear how the proposed iteration method performs relative to these stronger baselines”). Finally, reviewers raised unresolved clarity around iteration choice and convergence/accuracy, plus presentation issues (Reviewer iyWW: “numerous typos”). The rebuttal improves practicality and adds evidence, but some concerns remain sufficiently material to keep the paper borderline.

**Reviewer Concerns:**

- Addressed by the rebuttal: The rebuttal directly targets the “complexity” and tuning critiques (Reviewer iyWW: “adds considerable complexity… new hyperparameters… not trivial to choose”; Reviewer jKzN: “How can we determine T…?”; Reviewer gFyH: “How large should T be?”) by (a) quantifying IGM/CRM computation cost and arguing stability across categories, and (b) providing an automated procedure to select [Stack-GS-J-Else] parameters using a reconstruction MAE criterion. It also responds to requests for alternative solvers (Reviewer iyWW: “Did you try Anderson… SOR… block-Newton?”) by giving feasibility arguments and adding SOR experiments. It partially addresses scaling questions (Reviewer iyWW: “how does the method scale… memory or computation trade-offs?”) with a timing-based scaling study, and acknowledges the writing issues (Reviewer iyWW: “numerous typos”).

- Outstanding: Comparative positioning remains weak relative to the concern that the paper “does not compare against alternative approaches…” (Reviewer gFyH). The generality critique is only partially resolved: while the rebuttal adds evidence beyond TarFlow, it does not fully settle the concern that the method is “closely tied” / “tailored” to a narrow class of architectures (Reviewers gFyH, jKzN), nor does it fully eliminate the caveat that speedup is “not guaranteed” across AR flow designs (Reviewer iyWW). The scalability evidence is suggestive, but it does not fully answer how convergence, stability, and sample quality behave in trained high-resolution settings, which was the thrust of the scaling concern (Reviewer iyWW). The theory discussion still needs careful tightening to convincingly close the loop on “How large should T be?” and convergence-rate/approximation-accuracy requests (Reviewers gFyH, jKzN).

**Reviewer Scores:**

- Reviewer iyWW (original 6): Likely increases to 7. The main negatives were implementation, tuning burden, and lack of a clear hyperparameter recipe (“adds considerable complexity… new hyperparameters… not trivial to choose”; “paper does not provide a simple recipe”), which the rebuttal addresses with quantified metric cost and an automatic tuning procedure; remaining generality caveats are less central to their stated emphasis.

- Reviewer jKzN (original 4): Likely increases to 6. Their key objection was potential slowdowns when iteration counts are large (“When T is greater than the number of patches, the method will be slower…”) and missing guidance on T (“analysis… relationship between T and the image size”; “How can we determine T”), which the rebuttal addresses by emphasizing tough-block localization and automated parameter selection.

- Reviewer gFyH (original 4): Likely unchanged. Their main negatives were missing strong comparisons (“does not compare against alternative approaches…”) and insufficient convergence/accuracy guidance (“How large should T be? … convergence rate…”). The rebuttal improves parts of this, but the comparative gap and the need for sharper, fully convincing theory still leave a material portion of their critique outstanding.

---

### Decision · Program_Chairs · 2026-01-26

Reject